# Formation of a single quasicrystal upon collision of multiple grains

Insung Han [1,7,8], Kelly L. Wang [2,8], Andrew T. Cadotte[3], Zhucong Xi[1], Hadi Parsamehr[1], Xianghui Xiao [4], Sharon C. Glotzer [1,2,3,5,6✉] & Ashwin J. Shahani [1,5✉]

Quasicrystals exhibit long-range order but lack translational symmetry. When grown as single crystals, they possess distinctive and unusual properties owing to the absence of grain boundaries. Unfortunately, conventional methods such as bulk crystal growth or thin film deposition only allow us to synthesize either polycrystalline quasicrystals or quasicrystals that are at most a few centimeters in size. Here, we reveal through real-time and 3D imaging the formation of a single decagonal quasicrystal arising from a hard collision between multiple growing quasicrystals in an Al-Co-Ni liquid. Through corresponding molecular dynamics simulations, we examine the underlying kinetics of quasicrystal coalescence and investigate the effects of initial misorientation between the growing quasicrystalline grains on the formation of grain boundaries. At small misorientation, coalescence occurs following rigid rotation that is facilitated by phasons. Our joint experimental-computational discovery paves the way toward fabrication of single, large-scale quasicrystals for novel applications.

[1] Department of Materials Science and Engineering, University of Michigan, Ann Arbor, MI 48109, USA. [2] Macromolecular Science and Engineering Program, University of Michigan, Ann Arbor 48109 MI, USA. [3] Applied Physics Program, University of Michigan, Ann Arbor 48109 MI, USA. [4] National Synchrotron Light Source-II, Brookhaven National Laboratory, Upton, NY 11973, USA. [5] Department of Chemical Engineering, University of Michigan, Ann Arbor, MI 48109, USA. [6] Biointerfaces Institute University of Michigan, Ann Arbor, MI 48109, USA. [7] Present address: Department of Materials, University of Oxford, Parks Road, Oxford OX 13PH, UK. [8] These authors contributed equally: Insung Han, Kelly L. Wang. ✉email: sglotzer@umich.edu; shahani@umich.edu

The growth mechanism of quasicrystals (QCs) has attracted great interest[1,2] due to their unique crystal structure that cannot be explained by the presence of a unit cell[3]. Instead, QC lattices may be described by two or more space-filling motifs[4,5], or *tiles*. The presence of multiple tiles is due to an additional degree of freedom present in QCs called a *phason*[6], which encompasses associated modes of phason strain and its relaxation (e.g., phason flips)[6–8]. Phason strain describes the deviation from ideal quasiperiodicity[7] and phason flips are characterized by discrete particle motions[8] that result in a change in tiling configuration. These changes in tiling configurations are representative of phason excitations and relaxations, and do not introduce defects into the crystal[6]. Previous studies suggest that entropic contributions can have a significant influence on QC stability[1,2,9]. This suggests there is degeneracy in tile configurations that preserves quasiperiodicity and that QCs can grow into low or zero phason strain structures without the need for complex phason flip sequences[9].

Although past studies on QC growth mechanisms extend our understanding of phason contributions to stability in a bulk QC[1,2,9], studies on phason contributions to the formation and motion of grain boundaries (GBs) remain limited[10]. Yet the latter is critically important from a practical standpoint since the formation of polycrystals is unavoidable due to finite nucleation rates below the melting point. That is, two solid nuclei may impinge on one another, retaining their structure and forming a long-lasting GB. The GBs can, in turn, deteriorate material properties. For instance, defects in quasicrystalline films render the substrate underneath more vulnerable to corrosion by providing a channel for electrolytic attack[11]. In addition, the anisotropic thermal, electrical, and frictional properties of decagonal (d-)QCs can be maximized for future applications[12,13], provided the QCs are devoid of GBs between mismatched solid crystal nuclei.

Recently, Schmiedeberg et al.[10] conducted phase field crystal simulations to determine whether and when a GB may form between two QCs. In general, they observe that grain coalescence occurs more readily in dodecagonal QCs than in periodic crystals. This finding is remarkable given that two QCs are always incommensurate. They also reported that phasons play a significant role in distributing stress[10] around the non-fitting structures, where the resulting phason strain can be relaxed via phason flips[8]. Despite the insights obtained by Schmiedeberg et al.[10], the fundamental question remains: Can coalescence take place in experimental systems? If so, how can simulations support experimental observations?

Here, we investigate the growth dynamics of multiple thermodynamically stable d-QC grains[14] upon solidification of an $Al_{79}Co_6Ni_{15}$ alloy. We used synchrotron-based, four-dimensional (i.e., 3D space plus time-resolved), X-ray tomography (XRT) to resolve *grain coalescence*, or the time-dependent formation of a single d-QC from multiple grains. This experimental technique has opened a paradigm shift in solidification science, allowing us to capture transient microstructural dynamics in optically opaque materials[15–17]. To the best of our knowledge, our in situ experiments provide the first-ever demonstration of QC coalescence. On the basis of our experimental results, we performed molecular dynamics (MD) simulations to identify the mechanism of grain coalescence (or conversely, GB formation) at the atomic scale. We examine relevant crystal rotation mechanisms[18–22] and discuss the role of phasons in facilitating grain coalescence as well as their density as a function of misorientation between grains. Our combined efforts provide direct evidence of single crystal formation between incommensurate structures, such as QCs.

## Results and discussion

Figure 1 depicts the time-evolution of multiple d-QCs before and after collisions in an alloy of composition $Al_{79}Co_6Ni_{15}$, upon slow cooling (1 °C/min) from above the liquidus (~ 1026 °C) to below. The growth sequences of the as-grown d-QCs were recorded via XRT every 10 mins with 20 s scan time, starting from 1020 °C (at which point the sample was in a fully liquid state). The key advantage of using XRT is that we can unambiguously visualize the morphologies, misorientations, and growth dynamics of the QCs in real-time and in 3D, without needing to repeatedly quench our specimen. Quenching is known to distort the shapes and orientations of the solid–liquid interfaces[23]. We confirmed the existence of d-QCs in our sample through X-ray diffraction (XRD, see Fig. 2(a)) together with a thermodynamic assessment of the Al-Co-Ni system[14]. On the basis of our XRD results on a water-quenched specimen, we plot in Fig. 2(b) the physical scattering vector, $G_{||}$ (defined as $4\pi \sin\theta/\lambda$, where $\lambda$ is X-ray wavelength), against the full width at half maximum (FWHM) of the peaks belonging to the d-QC in Fig. 2(a). The linear relationship between the two quantities indicates that the contribution of phonon strain in the lattice dominates that of phason strain[24]. Therefore, we can expect that phason strain is relaxed if we consider the relatively longer time-scales of the slow-cooling experiment showcased in Fig. 1.

The d-QCs in Fig. 1 show a decaprismatic morphology[25] with a "long axis" parallel to ⟨00001⟩, representing the fast-growing periodic direction. Perpendicular to this direction is the aperiodic plane {00001}. Similar to our past experiments[25,26], the d-QCs are "anchored" to the oxide skin of the sample (not pictured), which acts as a fixed, heterogeneous nucleant for the d-QCs, preventing their displacement. Grain displacement would manifest itself as negative solid–liquid interfacial velocities on one side of the grain and positive interfacial velocities on the other. We do not observe this feature here nor in our prior experiments[25]. As the d-QCs grow, they interact with each other through *soft impingements* (overlapping diffusion fields) and *hard impingements* (collisions)[27].

We selectively focus on two different cases of hard collisions between d-QCs with (i) parallel long axes (Fig. 1(a–c)) and (ii) non-parallel long axes (Fig. 1(d–f)). Figure 1(a, d) shows these two cases after 50 min of continuous cooling from a viewpoint perpendicular to the long axes of the d-QCs. Figure 1(b, e) displays a bird's eye (or cross-sectional) view of the growth sequences from the quasiperiodic planes. When the long axes of d-QCs are parallel (Fig. 1(b)), we observed multiple coalescence events, the first between 20 and 30 min and the second between 50 and 60 min, the culmination of which is the formation of a single d-QC. The formation of a single d-QC is further evidenced by the absence of a GB groove (where the GB intersects the solid–liquid interfaces)[28,29], as well as the presence of ten facets on the coalesced structure at the final time-steps; this end-state is depicted in the outermost isochrone in Fig. 1(b) and the near-perfect symmetry of its facets is quantified via interface normal distribution[25,30] in Fig. 1(c). That is, GBs may be detected in solidification experiments[31–34] owing to the fact that they create macroscopic depressions (grooves) of the solid–liquid interface around the point at which they emerge into the liquid. According to Young's law[35], the groove angle, $\phi$, is related to the grain-boundary energy, $\gamma_{gb}$, as $\gamma_{gb} = 2\gamma_{sl}\cos(\phi/2)$, where $\gamma_{sl}$ is the solid–liquid interfacial free energy. If the GB groove persists during solidification, it can be inferred that the GB is stable and fixed to the groove. However, the morphological transition from a V-shaped groove to a faceted interface ($\phi \to 0°$, see thick arrow in Fig. 1(b)) suggests otherwise, i.e., the annihilation of the GB during grain coalescence. Interestingly, the facet orientations of the d-QCs prior to impingement were nearly the same as those d-QCs following coalescence. This observation is in line with the findings of Schmiedeberg et al.[10], who testify to the coalescence between two colloidal QCs with small initial misorientation in the

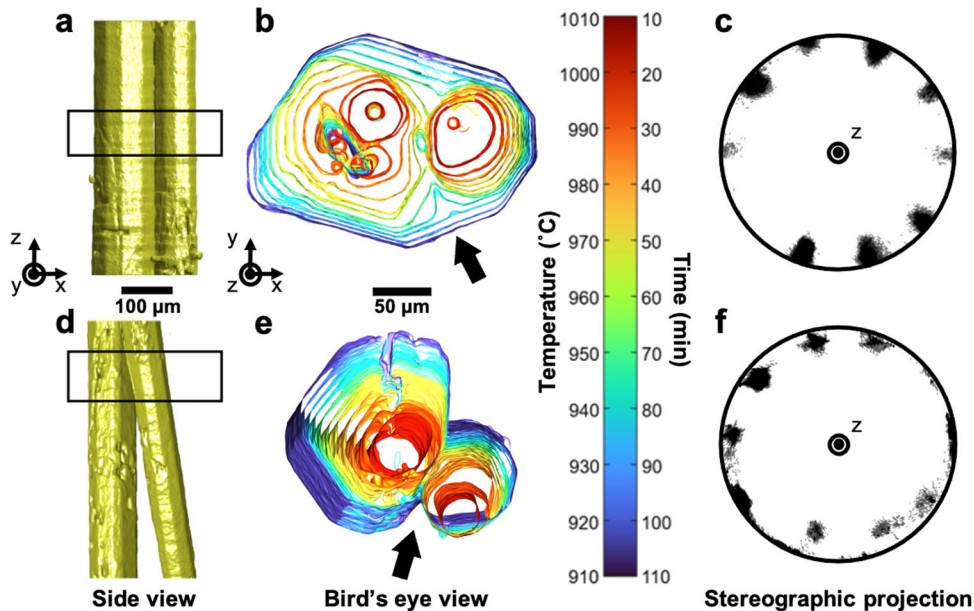

**Fig. 1 Tracking grain impingements in real-time. a** Side view ($\hat{z} - \hat{x}$ in the specimen frame) of two d-QCs with parallel $\langle 00001 \rangle$ long axes, observed after 50 min of cooling (1 °C/min) from 1020 °C. **b** Birds-eye view ($\hat{x} - \hat{y}$) of quasiperiodic plane corresponding to boxed region shown in **a**. **c** Stereographic projection of interface (facet) normal vectors of the outermost isochrone (910 °C) in **b**. Zone axis is $\hat{z}$. Facets are evenly projected (~ 36° apart) as peaks along the circumference (or primitive circle) of the projection plane, which implies the formation of a single, decaprismatic QC along $\hat{z}$. The missing tenth peak can be attributed to the less developed facet in **b**. **d** Side view of d-QCs with non-parallel $\langle 00001 \rangle$ long axes, observed at the same timestep as in **a**. **e** Birds-eye view of the boxed region shown in **d**. **f** Stereographic projection of interface normal vectors of the outermost isochrone (910 °C) in **e**. Zone axis is again $\hat{z}$. Quasiperiodic facets of two QCs are inclined to the projection plane, and thus the corresponding peaks do not lie on the primitive circle. Peaks are not always separated by 36° either, suggesting a superposition of two single crystal patterns and hence a bi-quasicrystalline structure. Isochrones of the solid–liquid interface in **b** and **e** are colored to illustrate the passage of time, with early times in red and late times in blue. Times and temperatures in **b** and **d** are as follows: 10 min (1010 °C), 20 min (1000 °C), 30 min (990 °C), 40 min (980 °C), 50 min (970 °C), 60 min (960 °C), 70 min (950 °C), 80 min (940 °C), 90 min (930 °C), 100 min (920 °C) and 110 min (910 °C). Thick black arrows in **b** and **e** point to the evolution of the grain-boundary groove in time.

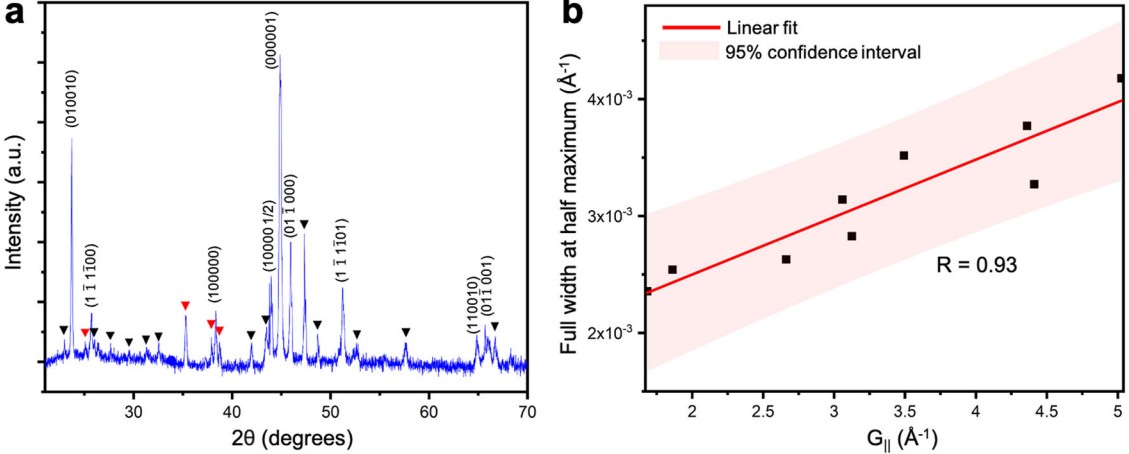

**Fig. 2 X-ray diffraction analysis. a** X-ray diffraction pattern of water-quenched $Al_{79}Co_6Ni_{15}$ alloy from an initial temperature of 1030 °C. Rapid quenching prevents peritectic transformation of d-QCs from proceeding to completion. The diffraction peaks are indexed to confirm the presence of decagonal (d-) QCs at 970 °C. The peaks indexed with black and red correspond to the $Al_3Ni$ and aluminum oxide, respectively. **b** A linear fit with Pearson correlation coefficient of 0.93 is shown between $G_\parallel$ and FWHM along with 95% confidence interval bounds, see text for details.

aperiodic plane regardless of the initial distance between them. Our quantitative analysis of facet orientations as a function of time (Supplementary Fig. 1) provides further support of this behavior. The groove itself does not move laterally along the solid–liquid interfaces on the intermediate time-scales (see arrow in Fig. 1(b)), which would indicate that GB translation on its own is not the mechanism of coalescence. The irregular shape of the

d-QC on the left-hand-side in Fig. 1(b) between 40 and 50 min of cooling (i.e., prior to collision) can be attributed to a mutual interference of diffusion fields between the two grains.

On the other hand, in the case where the two long axes are non-parallel to each other, we observed the persistence of a V-shaped GB groove (Fig. 1(d, e)), signifying the formation of a stable GB between the two grains. In comparison to the above

scenario where the long axes are parallel, here the quasiperiodic lattice in one d-QC merged with the periodic lattice of the other d-QC. Naturally, the appreciable mismatch between the two lattices resulted in the formation of a GB, and hence a V-shaped groove in Fig. 1(e). Quantitative support comes from the stereographic projection of interface orientations shown in Fig. 1(f), which represents here the superposition of two single QC patterns (cf. Fig. 1(c)). Since contrast in XRT stems from differences in photoabsorption between the phases, we can only capture the external solid–liquid interfaces and not the GBs and atomic configurations within the solid phases. Furthermore, it is nearly impossible to preserve the QC interfaces at room temperature for high-resolution imaging, since the decagonal phase undergoes a peritectic decomposition below 910 °C[14]. Thus, to overcome such experimental constraints, we turn to MD simulations to "see" inside the evolving d-QCs and fill in the spatiotemporal gaps from our XRT experiment.

We systematically examine the effects of misorientation on grain behavior in d-QCs using seeded MD simulations with an isotropic, single-component pair potential[36]. We focus on misorientations within the aperiodic plane {00001}, since the XRT experiment provides evidence for coalescence when the long axes are parallel (Fig. 1(a–c)). Thus, we carry out MD simulations in quasi-2D boxes to maximize surface area along the quasiperiodic plane. We fix the initial position of d-QC seeds in our simulations to match experimental conditions, where grains were "anchored" to the sample surfaces. Seed positions are fixed for the entire simulation. We use the decatic order parameter, $\psi_{10}$, to determine the alignment of the local bond configurations about each particle. We will call this quantity the local grain orientations (LGO) and particles with local bond configurations that align with the reference basis will have a LGO of $\theta = 0°$. A detailed description of simulation setup and analysis is provided in the Methods section.

To elucidate the mechanism behind grain coalescence in d-QCs, we begin with characterization of GB formation as a function of misorientation. Here, we define misorientation as the rotation angle about the {00001} axis between two seeds. We define *grain* as a region where the arrangement of particles may be described by a continuous lattice in physical space and *seed* as a set of particles belonging to the d-QC lattice with fixed LGOs. We will refer to simulations based on the misorientations between seeds, rather than the misorientations between grains, since grains can change LGOs over time but the misorientations between seeds are fixed and represent the initial conditions of d-QC grains. Figure 3 compares small (3°), intermediate (9° and 10°), and large (15°) seed misorientations between two d-QC seeds with a fixed distance of $L = 40d$, where $d = 1.02$ is defined as the distance between $r = 0$ and the 1st neighbor shell in the isotropic pair potential[36] and each seed contains 144 atoms. Each seed accounts for ~ 0.03 % of the particles in the entire simulation. This means that the contribution of the fixed seed to the phason dynamics around the seed could be negligible. We verify this assumption with analysis of particle flips in both fixed and unfixed seed simulations (Supplementary Fig. 2). No significant suppression of phason dynamics was observed around the seeds in both cases. In unfixed seed simulations, we do not see grain displacement upon collision (Supplementary Fig. 3), which suggests our use of fixed seeds does not artificially hinder grain motion and should not introduce any significant artifacts into our results. Therefore, our choice to incorporate fixed-seed simulations can be justified by the purpose of replicating solidification experiments. Temperature was fixed at $kT = 0.5$ reduced units and pressure was fixed at $P = 3.9$ reduced units. Further description of reduced units are provided in the "Methods" section.

When seed misorientations are 3° and 9° (Fig. 3(a, b)), we observe unimodal histograms of $\theta$ in the bulk QC at equilibrium (left column) and a ten-fold pattern in the diffraction (right column). These results suggest that the misoriented grains rotated and the misorientation was minimized. On the other hand, when seed misorientations are 10° and 15° (Fig. 3(c, d)), we observe bimodal histograms of $\theta$ (left column) in the bulk QC at equilibrium and overlapped, ten-fold patterns in the diffraction image (right column). These results indicate the formation of a GB between misoriented grains. In the case of Fig. 3(a), one grain (yellow-orange) adopts the LGOs of the other grain (blue), see middle column. For intermediate seed misorientations (Fig. 3(b, c)), grains rotate toward intermediate orientations (dark colored regions, middle column). In some cases, a GB is observed because misorientation is reduced, but not eliminated (Fig. 3(c)). This is visible in the diffraction pattern (right column), where the misorientation between the two ten-fold patterns is ~6° after rotation. At large seed misorientation, i.e., 15°, the GB is clearly defined and orientations of both seeds strongly resemble the initial seed orientations (Fig. 3(d)). The diffraction pattern in Fig. 3(d) shows a misorientation of ~ 15°, which corresponds to the initial misorientation between the seeds (i.e., no grain rotation). Note the diffraction patterns shown here are closely related to the stereographic projections of facet orientations from experiment (Fig. 1(c, f)). This is because the external morphology of a crystal reflects its point group isogonal with its space group[37].

In light of these trends, we can identify a critical seed misorientation, $\Delta\theta_{crit} \approx 9.5°$, for the given simulation conditions ($L = 40d$ and $kT = 0.4$), below which grains can rotate. The angle of critical value provided here is not meant to represent all cases of grain coalescence in d-QCs, as $\Delta\theta_{crit}$ is likely a function of various thermophysical parameters (i.e., temperature, grain size, fluid viscosity, and external stress[18,19]). Instead, we treat it as a reference point for how the behavior of d-QCs change at $\Delta\theta_{crit}$. After grain rotation towards 0° misorientation from the small initial misorientation (3°), only a few dislocations in the GB region were detected (Fig. 4(a, c)) in the single density mode[38], which suggests grain coalescence. In contrast, we observed an array of dislocations in the single density mode[38] from the simulations with 15° initial misorientation ($> \Delta\theta_{crit}$), indicating the formation of a GB (Fig. 4(b, c)). That is, grain coalescence occurs only for relatively small seed misorientation angles, see Fig. 4(c). We also verify grain coalescence and formation of GBs using density modes (Supplementary Fig. 4), and tiling analyses (Supplementary Fig. 5).

We begin our analysis of coalescence by mapping the LGOs of grains grown from two seeds with misorientation 6°, well below $\Delta\theta_{crit}$, as depicted in Fig. 5. Figure 5(a–e) shows the time evolution of coarse-grained LGOs for seeds with 6° misorientation. At early timesteps (Fig. 5(a, b)), we observe two grains (labeled A and B) with good alignment to seed orientation (yellow-orange and blue regions, seeds A and B, respectively). Immediately after collision, grain A remains well aligned with seed A and a GB is clearly visible (Fig. 5(b)). Both grains continue to rotate and reduce misorientation (darkening of both grains, Fig. 5(b–e)) over the next ~ $4.33 \times 10^6$ timesteps. The GB grooves (Fig. 5(b, c), arrows) gradually flatten during rotation. This confirms the formation of a single grain, mimicking the experimental results (Fig. 1(b)).

For comparison to our d-QC system in Figs. 3 and 5, we show results from a FCC simulation with seeds at a fixed distance of 15.5a, where $a = 4\frac{d}{\sqrt{2}}$ is the lattice constant, $d = 1.1$ is average bond length in the FCC grain, and 3° misorientation between seeds. The results are shown in Fig. 6, where the image is colored using the same method used in Figs. 3 and 5, where alignment with seeds A and B (labeled in Fig. 6(a)) are indicated by a yellow-

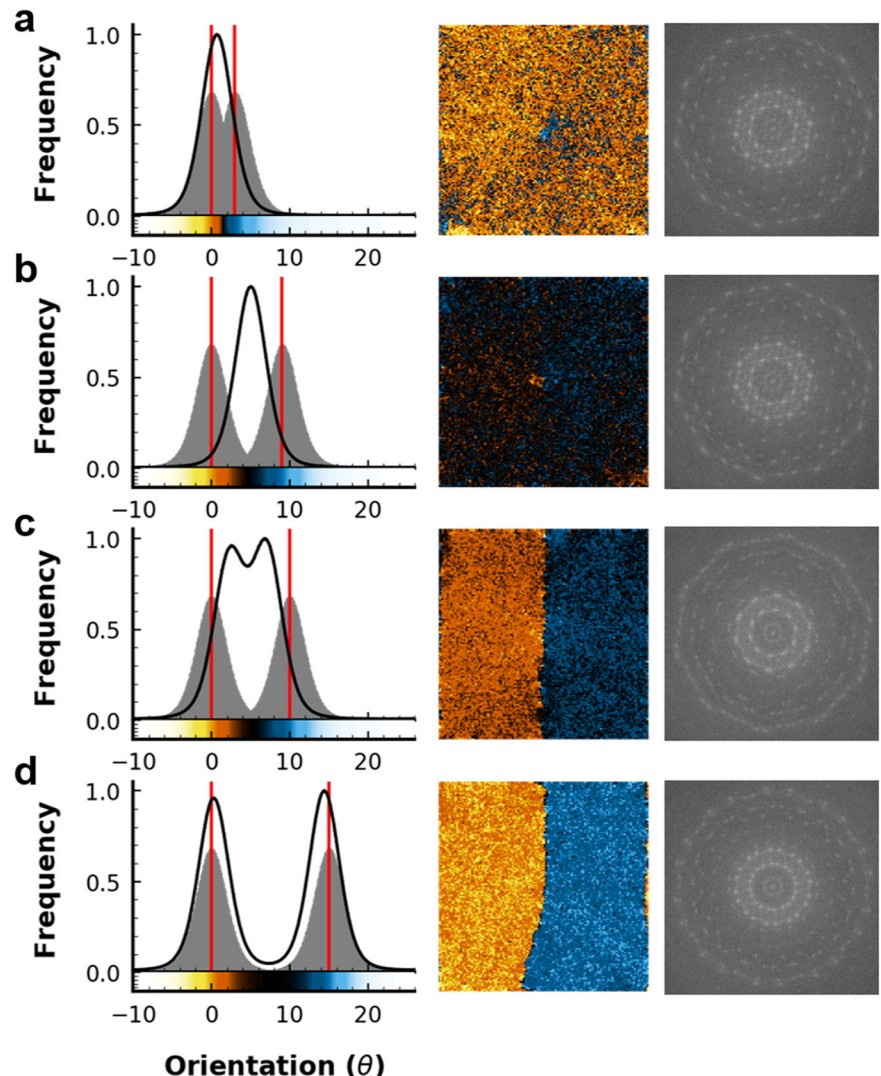

**Fig. 3 Simulation orientation analysis.** MD simulations showing changes in local grain orientations (LGOs,$\theta$) towards near-equilibrium configurations (~ 25 million simulation timesteps). Left column: Histograms of simulated (black line) and expected seed LGO distributions (gray peaks). Red lines indicate the expected LGOs of particles in each seed. Histogram bin frequencies (arbitrary units) for simulation lattice orientations were averaged after $\theta$ reached a constant value. The thickness of the black line represents the standard deviation of frequency for each bin. Gray peaks are expected probability density functions (PDF) for the reference grain ($\theta = 0°$) and rotated grain. PDFs are calculated from single-seeded simulations. Peaks for the reference seed are centered at $\theta = 0°$ and peaks for the rotated seed are centered at $\theta =$ (**a**) 3°, (**b**) 9°, (**c**) 10°, and (**d**) 15°, respectively. Middle column: Spatially binned simulation frames at 25 million timesteps for each set of seeds. Here, coarse-grained images are shown, rather than images with atomic level resolution, because they show grain rotation and GB characteristics more clearly. One pixel or cell represents 20 particles on average. All images show the aperiodic {00001} plane of our d-QC simulations. Colorbars for $\theta$ are below each histogram (left) and correspond to the orientation in the histogram axes, where yellow-orange corresponds to particles that align with the reference seed ($\theta = 0°$). Bright blue corresponds to particles that align with the rotated seed. Dark regions on the colorbar correspond to angles along the shortest arc between 0° and the rotated seed and white regions on the colorbar indicate angles along the longest arc between 0° and the rotated seed. White regions typically correspond to liquid regions, which are not visible in fully crystallized simulations frames (middle column), but are visible during growth, when liquid is still present in simulation. Right column: Calculated diffraction patterns based on the atomic level resolution simulations used in the middle column. Note the diffraction patterns of **a** and **b** are indicative of a single d-QC. On the other hand, the diffraction patterns of **c** and **d** suggest the presence of two d-QCs with different orientations.

orange and blue color, respectively, and fluid-like regions are represented by white. In contrast to our d-QC results, simulations of FCC grains (Fig. 6) show clear GB grooves at all stages of growth and more localized changes in misorientation (Fig. 6(b–e)). We note that, even after coalescence, small regions remain well aligned with the initial d-QC seeds (Fig. 3(a, b) and 5(e)), despite rotation through the bulk of the crystal. This is due to our decision to fix the seed position throughout the entire simulation, preventing rotation of the nucleation sites of each grain.

The phenomenon of grain coalescence is well documented in polycrystalline materials[18,19,21,39]. Mechanisms for grain coalescence are driven by reduction of GB free energy and are categorized broadly by GB migration and grain rotation, as alluded to earlier. In embedded grains, GB migration is typically coupled with rigid sliding[18,19,39,40] (termed *coupling*), which is identified by a translation of the GB and a concomitant, continuous change in misorientation (i.e., lattice rotation) near the GB[18]. In contrast, uncoupled GB sliding manifests as a uniform, global change in grain orientation, as we observed in Figs. 5–6. Measurements of

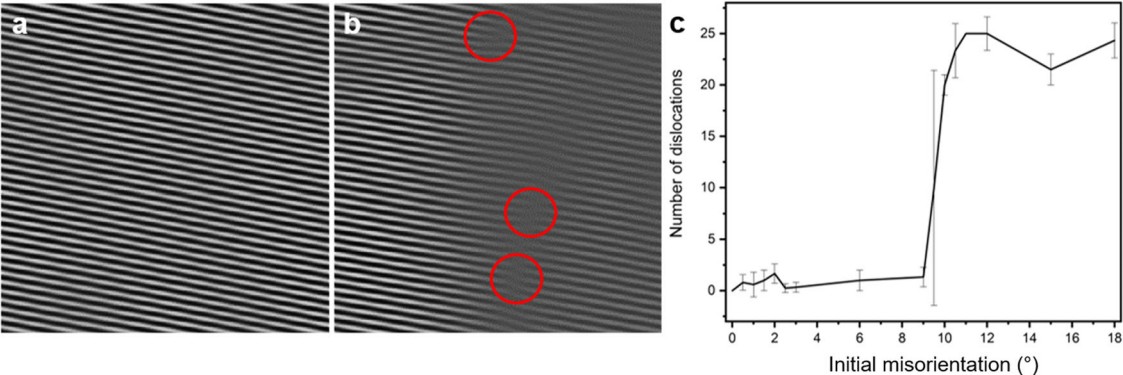

**Fig. 4 Number of dislocations along the grain boundary.** Real-space images of a single density mode[38], in the region where QCs collide, taken from the last frame in MD simulation (~ $2.5 \times 10^7$ simulation timesteps) with **a** 3° and **b** 15° initial misorientations. Dislocations are highlighted with red circles. **a** and **b** are cropped to provide a magnified view from the images that represent full volume. **c** Relationship between the initial misorientation and number of dislocations along the grain boundary in the coalesced structure. We find that there are few dislocations (if any) at low ($\theta \lesssim 9.5°$) initial misorientation, since two QCs can rotate toward $\theta = 0°$ (cf. **a**). Conversely, there are many dislocations when two QCs cannot minimize the misorientation between them. The error bars were calculated from multiple dislocation analyses to retain consistency of our approach.

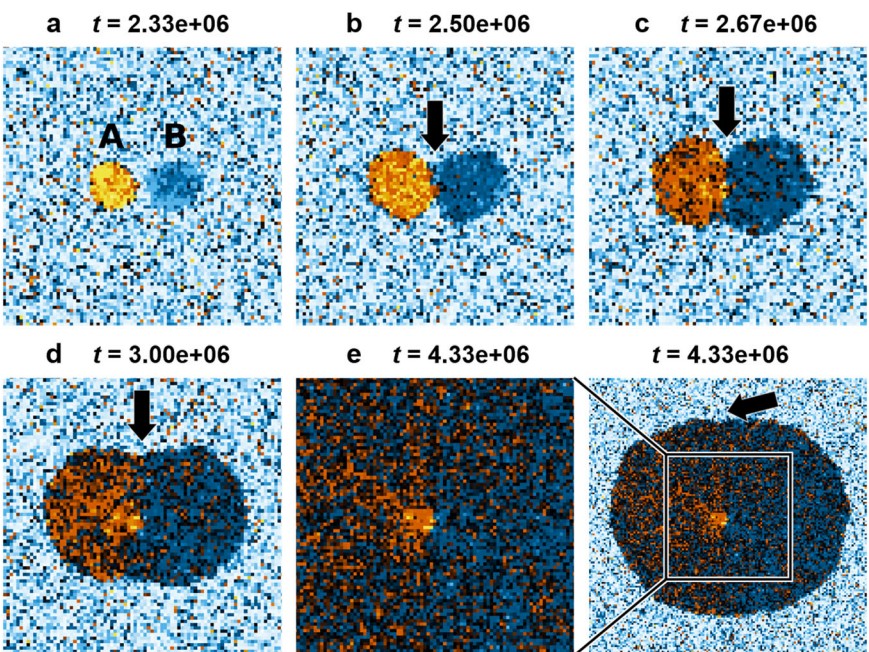

**Fig. 5 Growth of a single d-QC from two seeds.** Seeds are labeled A and B in **a** with 6° misorientation. Here, coarse-grained images of the aperiodic {00001} plane show the local grain orientations (LGO) of d-QC grains **a** during early stages of grain growth; **b** immediately after collision; **c** after collision, grain rotation to minimize grain misorientation; **d** early stages of grain coalescence; **e** after grain coalescence. All heatmaps are cropped according to **e** from the total volume. Subplots **a**–**e** are colored on the basis of an expected particle seed distribution, as shown in the middle column of Fig. 3. Contiguous regions of white in subplots **a**–**e** correspond to liquid regions. Arrows in **b** and **c** point to the GB grooves at the QC-liquid interfaces. Arrows in **d** and **e** point to regions where the GB groove has flattened.

grain rotation rates as a function of initial grain misorientation (Supplementary Fig. 6) show strong and qualitative agreement with classical equations for the dependence of GB free energy ($\gamma_{gb}$) on misorientation ($\theta$)[41] and driving pressure of sliding, $P_{\parallel} \approx -\frac{d\gamma_{gb}}{d\theta}\frac{1}{\mathscr{R}}$, where $\mathscr{R}$ denotes grain radius[18]. The presence of sliding and the gradual disappearance of the GB groove in Fig. 5 suggests d-QCs are able to eliminate GBs with pure rotation. This is unexpected—due to the aperiodic, long-range translational order present in QC lattices, translational mismatch between lattices grown from two seeds is likely, even when misorientation between grains is 0°. It should not be possible to eliminate translational mismatches with rotation alone. This is consistent with our simulations of FCC grains (Fig. 6), wherein the GB

groove remains stationary and prominent, despite rotation towards 0° misorientation between grains. This finding also means that, although GBs are not observed in the physical space of our d-QCs, evidence of translational mismatches may be found in the phason space of the crystal.

We note a few caveats to consider when applying this theoretical framework to experiment. First of all, since the sliding velocity is directly proportional to the driving pressure, $P_{\parallel}$, and $P_{\parallel}$ is itself inversely proportional to the radius, $\mathscr{R}$, of the grain[18], a smaller $\Delta\theta_{crit}$ and slower rotation rate are expected in the XRT experiment, compared with the simulation. Second, in experiment, grains are anchored onto a substrate. Even so, grains may rotate if the applied torque is sufficient to overcome any

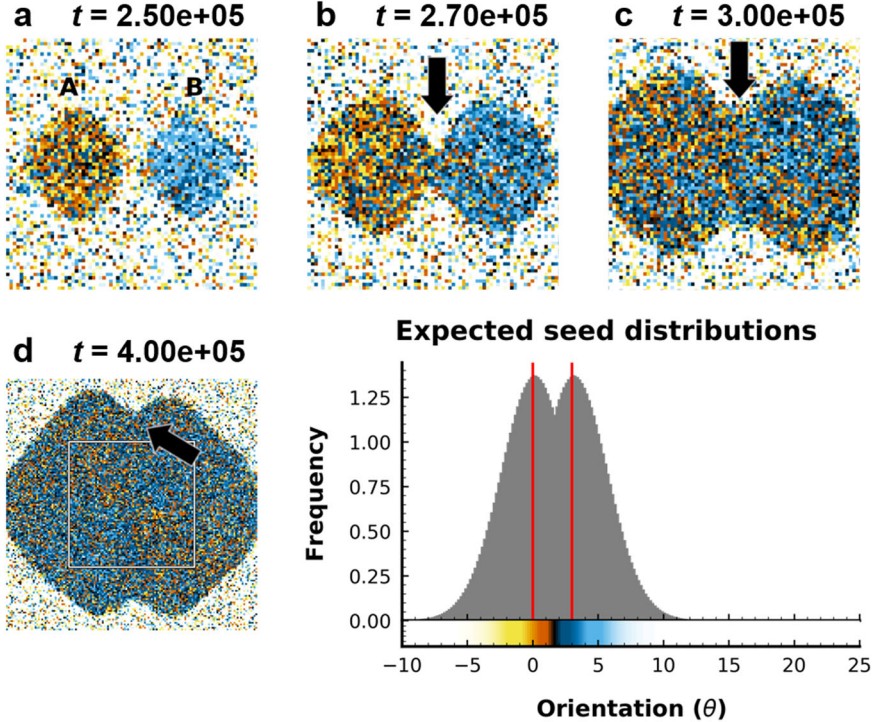

**Fig. 6 Growth of two FCC grains with 3° misorientation.** Growth at **a** 2.50 × 10⁵, **b** 2.70 × 10⁵, **c** 3.00 × 10⁵, **d** 4.00 × 10⁵ timesteps. **a–c** Frames from early stages of growth are cropped to show more detail. The area cropped is shown by the white outline in **d**. **e** shows the colorbar for this figure, where alignment with seed A corresponds to yellow-orange and alignment with seed B corresponds to blue color. Here, the gray peaks represent the expected probability density functions (PDFs) for reference grain ($\theta = 0°$) and the misoriented grain ($\theta = 3°$). The colorbar is cropped from the maximum range of orientation (–10° to ~ 80°) for clarity. Black area indicates angles along the shortest arc between 0° and the misoriented seed, and white areas correspond to angles along the longest arc between 0° and the misoriented seed. GB grooves are present during all stages of growth for both orientations, despite rotation of both grains to 0° misorientation (see arrows in **b–d**). Although we observe global rotation of misoriented grains toward 0° misorientation, a persistent grain-boundary groove suggests unresolved phonon strain along the grain boundary due to incommensurate distances between FCC lattices.

interaction between grains or with an external substrate[42–44]. This suggests that the presence of external constraints may impact the kinetics of rotation; nevertheless, grain rotation is still possible. A case in point is Fig. 1(a, b) from our own work, showing grain rotation on experimental time-scales in the limit of small initial misorientation.

Owing to the unique symmetries of QCs, all dislocations in d-QCs contain phasonic components[8]. These phasonic components represent excitations in QC lattice; they contribute to an increase in the system's phason strain, which may be relaxed over diffusive timescales. In contrast, dislocations in periodic crystals are purely phononic. Previous work shows that motion of dislocations through d-QCs results in a wall of phasons along the slip plane[45]. This suggests that the dislocation reactions that enable sliding to occur (vide supra) redistribute the phonon strain from lattice mismatches as phason strain through a combination of dislocation annihilation mechanisms and phason flips. Owing to the complexity and variety of dislocations in d-QCs, however, we reserve quantitative analysis of phasonic motion and dislocation kinetics for future work. Instead, we examine simulations where grain coalescence occurs for residual phason strain as a potential signature of dislocation motion during grain coalescence (Fig. 7).

Density modes (Fig. 7(a–c)) are obtained by filtering two pairs of Bragg peaks from the diffraction image of the merged QC. They represent the two different length scales (e.g., golden ratio) in QCs. The region boxed in Fig. 7(a–c) with yellow rectangles is further processed for the phason strain analysis with an appropriate threshold (Fig. 7(d–f)). According to Freedman et al.[7], the "jags" in the stripes pattern are the signature of phason strain in

the quasiperiodic lattice. We quantified the fraction of "jags", which are longer than zero and shorter than the longer edge of the yellow rectangles in Fig. 7(a–c), along the direction of the stripes. Figure 7(g) shows that a higher phason strain is accumulated within the grain-boundary region as the initial misorientation increases in the simulation timescale, which supports the hypothesis that phonon strain is redistributed as phason strain during coalescence. This is in good agreement with the coalescence mechanisms shown in ref. [10], where coalescing grains were shown to transform from structures with phonon and phason strain, to a structure with phason strain, which is later relaxed.

Phason strain relaxation occurs over much longer timescales than our simulations[46]. This means that, although we expect d-QCs to gradually relax to a low or no phason strain state over time, phason strain introduced during grain coalescence should be observable. Thus, we expect phason strain relaxation to occur over experimental timescales near regions where the hard collision occurred (Fig. 7). Indeed, there is ample evidence of phason strain relaxation, obtained from recent in situ experiments. For example, Nagao et al.[47] observed grain-boundary migration in QCs through an "error-and-repair" process, wherein phason strain at the grain-boundary region is relaxed to generate an ideal quasicrsytalline lattice. In addition, we confirm the relaxation of phason strain in our slow-cooling experiments, as evidenced by the linear relationship between $G_{||}$ and FWHM of the peaks in the diffraction pattern (Fig. 2).

We elucidated the growth interaction between two d-QCs with fixed-seed positions via 4D XRT and MD simulation. To the best of our knowledge, this is the first experimental study to

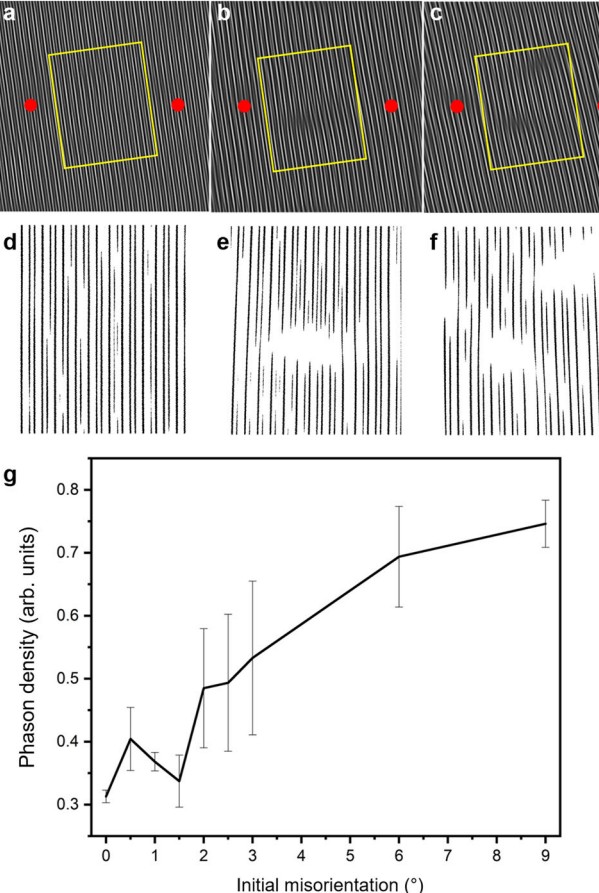

**Fig. 7 Phason density of coalesced structure.** Phason density modes obtained by filtering two pairs of Bragg peaks[7] from the merged quasicrystals with **a** 0°, **b** 3°, and **c** 9° of initial misorientations at timestep $4.0 \times 10^5$ from MD simulations. The red dots indicate the seed positions and the region highlighted with yellow rectangles is binarized for quantitative analysis of phason strain. **d**, **e**, and **f** correspond to **a**, **b**, and **c**, respectively. **g** Phason densities calculated as a function of initial misorientation between seeds, using the method introduced by Freedman et al.[7]. The phason density was determined from the binarized images based on the areas inside of the yellow rectangles for misorientations of multiple 0°, 0.5°, 1°, 1.5°, 2°, 2.5°, 3°, 6°, and 9° cases. For consistency of our results, we repeated analyses on different MD simulation datasets, which explains the origin of the error bars.

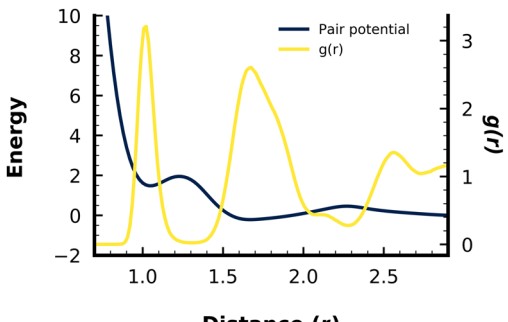

**Fig. 8 Tabulated pair potential and radial distribution function for quasicrystal simulations.** The single-component, isotropic pair potential (dark blue) used for all decagonal quasicrystal (d-QC) simulations is derived from the potential mean of force of a diamond-forming system[51]. The radial distribution function is shown in yellow, $g(r)$, for a single-seeded d-QC simulation after $19.7 \times 10^6$ timesteps.

investigate the structural continuity between two quasicrystalline crystals after a hard collision. With our joint analyses, we were able to provide a cohesive picture for the conditions that give rise to grain coalescence: (i) parallel periodic axes and (ii) small misorientation between quasiperiodic crystals. In this operating regime, we observed grain rotation toward 0° misorientation in order to minimize grain-boundary energy. This process occurs through a dislocation-mediated mechanism that allows the d-QCs to redistribute phonon strain due to lattice mismatch as phason strain, by local rearrangement of dislocations into valid tilings. Taken all together, our integrated approach highlights the exciting opportunity for microstructure optimization via control of the grain boundaries – that is, defect engineering. It provides the knowledge base for fabrication of defect-free QCs (e.g., through a controlled sintering process), thereby widening their potential uses and applications.

## Methods

**Synchrotron-based X-ray microtomography.** The 4D synchrotron-based X-ray microtomography (XRT) experiment was carried out at the 2-BM beamline of the Advanced Photon Source at Argonne National Laboratory in Lemont, Illinois, United States. A high-purity alloy ingot with nominal composition $Al_{79}Co_6Ni_{15}$ was prepared by the vacuum arc remelting (VAR) technique at the Materials Preparation Center (MPC) at Ames Laboratory in Ames, Iowa, United States. Then, the sample was cut into cylinders of 1 mm diameter by 5 mm height. The tomography scan was taken with a polychromatic "pink" beam centered at 27 keV every ten minutes as the sample was cooled from 1020 °C at a rate of 1 °C/min. Each scan consisted of 1000 projection images equally spaced in the range 0° to 180°, collected with an exposure time of 14 ms per projection. The projection FOV measured 2560 by 800 pixels, resulting in a pixel size of 0.65 μm².

The raw projection images were reconstructed into 800 2D image slices along the cylinder height by TomoPy[48], a Python-based open source framework developed for data processing and reconstruction. The processed 2D images show a clear absorption contrast between d-QCs and melt. This can be attributed to a non-congruent solidification of d-QCs, such that Al is rejected as growth proceeds[25]. The solidification pathway for an alloy of composition $Al_{79}Co_6Ni_{15}$ was verified by Thermo-calc[49] using the Al-Co-Ni thermodynamic database by Wang and Cacciamani[14]. For our further analysis, we segmented the solid–liquid interfaces with an appropriate threshold and the segmented images were combined to give the 3D microstructures. Volumes were rendered using the Image processing toolbox in Matlab R2018b.

**X-ray diffraction experiment.** A high-purity alloy ingot with nominal composition of $Al_{79}Co_6Ni_{15}$ was fully liquified at 1030 °C and slowly cooled (1 °C/min) to mimic the cooling profile of the XRT experiment. Then, the sample was water-quenched at 970 °C. X-ray diffraction pattern was obtained at room temperature using a Rigaku Smartlab diffractometer with a Cu-Kα radiation source.

**Decagonal quasicrystal simulation.** Molecular dynamics (MD) simulation was performed with HOOMD-blue[50] in the isobaric-isothermal (NPT) ensemble. Simulations used reduced units of energy ($\varepsilon$), length ($d = 1.02$) in arbitrary units, mass ($m$), and time ($\tau = \sqrt{\frac{md^2}{\varepsilon}}$). Particles interacted through an oscillatory, double-well potential[36] (Fig. 8), previously shown to form d-QCs,

$$V(r) = V_{\mathrm{PMF}}(r) - \varepsilon \exp\left(-\frac{(r-d)^2}{2\sigma^2}\right) - V(r_{\mathrm{cut}}) \qquad (1)$$

where $V_{\mathrm{PMF}}(r)$ is the tabulated potential of mean force (PMF) constructed from the radial distribution of a diamond-forming system[51], $\varepsilon = -1.8$, $d = 1.02$ is the location of the first minimum, $\sigma$ is the width of the Gaussian applied to the first well, and $r_{\mathrm{cut}} = 2.9d$ is the cutoff. The pair potential was smoothed and shifted to 0 at $r_{\mathrm{cut}}$.

Each simulation was performed with 500,000 particles and with periodic boundary conditions in three dimensions. Simulations were carried out in quasi-2D boxes with final average dimensions of $360 \times 360 \times 8$ to reduce the influence of layer mismatches and rearrangements along the periodic axis and to maximize the amount of atomic rearrangements in the quasicrystalline plane. Systems were linearly cooled from a liquid-like configuration ($T^*_{\mathrm{init}} = 1.5$) to a temperature near, but below the melting point ($T^*_{\mathrm{end}} = 0.4$) over 100,000 timesteps. The temperature was expressed in a reduced unit of $T^* = \frac{k_b T}{\varepsilon}$, where $k_b$ is the Boltzmann constant. Each system was then held at $T^*_{\mathrm{end}}$ for 20 million simulation timesteps at a pressure of 3.9. Simulations consisted of two fixed seeds with a distance of 40 between the seed centers and misorientation between 0° and 18°. For each misorientation, 3–5 simulations were run to ensure consistency of the results. The computational workflow and general data management in particular for this publication was primarily supported by the signac data management framework[52].

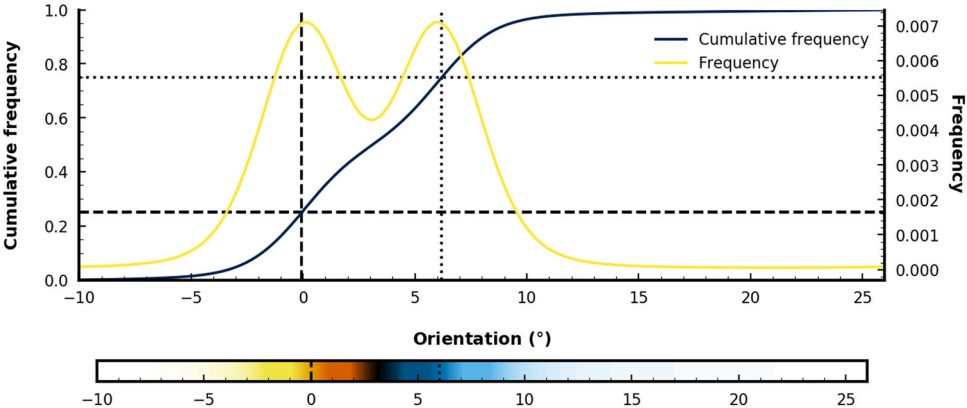

**Fig. 9 Orientational order mapping for quasicrystal simulations with 6° misorientation.** Mapping local grain orientations (LGOs) onto the cumulative histogram ($cf(\theta)$, blue line) of LGOs for 6° misorientation between seeds. Black dashed lines indicate the LGO when $cf(\theta_{0.25}) = 0.25$. Black dotted lines indicate the LGO when $cf(\theta_{0.75}) = 0.75$. These regions correspond to the peaks (means) in the expected bimodal histogram (yellow line, $f(\theta)$) of LGOs. Using values from the cumulative histogram, we create a colormap where alignment with the reference seed ($\theta_{0.25} = 0°$, dashed lines) corresponds to yellow-orange and alignment with the rotate seed ($\theta_{0.75} = 6°$ in this example, dotted lines) corresponds to blue. Black corresponds to LGOs between 0° and the LGO associated with the rotated seed (6° in this example), while white corresponds to $-10° < \theta < 0°$ or $16° < \theta < 28°$. Owing to the 10-fold rotational symmetry of the d-QC, the angle between each basis vector is 36°. We keep all LGOs values between $-10° < \theta \leq 26°$, where a value of 27° would be equivalent to $-9°$. This means that the white region could also be defined as an interval between $0° < \theta < \theta_{0.75}$, but the arc it represents is longer than the arc for the black region for all seed misorientations below 18°.

*Orientational order parameter analysis.* We used the bond-orientational order parameter for $k$-atic rotational symmetry[53] to identify the local orientational symmetry of each particle $m$:

$$\psi_k(m) = \frac{1}{n} \sum_j^n e^{ki\theta_{mj}} \qquad (2)$$

where $n$ is the number of neighboring particles and $\theta$ represents the angle between local bond orientations and a fixed basis with $k$-atic rotational symmetry, or, the local grain orientation (LGO). For a decagonal quasicrystal, we use the value of $k = 10$, so that $\psi_{10}(m)$ refers to the decatic order parameter. For FCC simulations, we computed $\psi_4(m)$, or the quadratic order parameter, on the 2D projection of the crystal down the 2-fold rotational axis. We note that the projected structure resembles a simple square lattice, resulting in fourfold, rather than twofold rotational symmetry in the 2D lattice. We determine neighbors for particle $m$ as the $k$ nearest neighbors to ensure that measurements reflect the local atomic environment of each particle. Data analysis for this publication utilized the freud library[54].

*FCC simulations.* We carried out additional MD simulations of an FCC forming system via the 6-12 Lennard-Jones pair potential,

$$V_{LJ} = 4\epsilon \left[ \left( \frac{\sigma}{r} \right)^{12} - \left( \frac{\sigma}{r} \right)^6 \right] \qquad (3)$$

that is truncated at $r_{cut} = 2.5 * \sigma$, where $\sigma = 1.0$, $\epsilon = 1.0$, and $r$ is the distance from the center of a particle, expessed in units of $\sigma$. Temperature was fixed at $kT = 0.6$, and pressure was fixed at $P = 1.0$ in reduced units. All periodic simulations contained $\approx 500,000$ atoms and contained either one or two fixed seeds, with 499 atoms each, and the potential was shifted and smoothed so that both the potential and its derivative are zero at the cutoff.

*Orientational order paramater mapping.* For qualitative comparison of grain rotation between two seeds with different misorientation, we normalize local grain orientations (LGOs) of individual particles based on their deviation from seed LGOs. To begin, we constructed expected histograms for two seeded simulations as bimodal distributions from histograms of single-seeded reference simulations (yellow line, Fig. 9), where peaks correspond to the reference seed misorientation ($\theta = 0°$) and the rotated seed misorientation ($0° \leq \theta \leq 18°$, 6° in Fig. 9). We remap the LGO of individual particles to the bimodal cumulative histogram, $cf(\theta)$. This highlights small changes in misorientation and makes it possible to compare grain interactions at various misorientations without manual rescaling.

This is because the cumulative distribution $cf(\theta)$ always equals 0.5 when $\theta$ equals the mean of a unimodal distribution. If we construct our bimodal distribution as a mixture of two Gaussian-like distributions with identical variance, but different means, it follows that the means in the bimodal distribution correspond to $cf(\theta_{0.25}) \approx 0.25$ and $cf(\theta_{0.75}) \approx 0.75$, where $\theta_{0.25} = 0°$ and $\theta_{0.75}$ corresponds to the LGO of the rotated seed (in Fig. 9, $\theta_{0.75} = 6°$). Particles with LGOs different from either seed have $cf(\theta)$ of 1 or 0, and seeds with intermediate orientations have $cf(\theta)$ between 0.25 and 0.75, where LGOs with $cf(\theta)$s of 1 or 0 have the same meaning. This is because, for the 10-fold rotational symmetry found in decagonal quasicrystals, the basis vectors are invariant for rotations 36°, meaning, LGOs of $-10°$ and 26° are equivalent. Here, we define intermediate

orientations as angles that lie along the shortest rotation between two seeds (black region in Fig. 9) and unlike orientations as angles that lie along the longest rotation between two seeds (white region in Fig. 9). The inclusion of the extreme cases, where orientations are highly unlike either seeds, is useful in identifying liquid regions in the d-QC during growth, such as in Fig. 5. It is important to note that as misorientation approaches 18°, the difference between "intermediate" orientations and "unlike" orientations diminishes.

## Data availability

The datasets generated during and/or analyzed during the current study are available from the corresponding authors upon reasonable request.

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

## Acknowledgements

We are grateful to Pavel Shevshenko for assisting in the synchrotron-based tomography experiment. We thank Sophie Y. Lee and Bradley D. Dice for providing simulation seeding code and Joshua A. Anderson for valuable guidance on simulation set up. The work was supported by the US Department of Energy (DOE), Office of Science, Office of Basic Energy Sciences, under Award No. DE-SC0019118. This research used resources of the Advanced Photon Source, a U.S. Department of Energy (DOE) Office of Science User Facility operated for the DOE Office of Science by Argonne National Laboratory under Contract No. DE-AC02-06CH11357. Computational work used the Extreme Science and Engineering Discovery Environment (XSEDE)[55], which is supported by National Science Foundation grant number ACI-1548562; XSEDE award DMR 140129. Additional computational resources and services supported by Advanced Research Computing at the University of Michigan, Ann Arbor.

## Author contributions

I.H. and A.J.S. conducted the XRT experiment and analyzed the results. Beamline setup was prepared by X.X. K.L.W. and A.T.C. designed and ran the MD simulation with guidance from S.C.G. I.H., K.L.W. and Z.X. analyzed MD simulation results, with input from H.P. S.C.G. and A.J.S. planned and supervised the project. All authors discussed the results and contributed to the writing of the manuscript.

## Competing interests

The authors declare no competing interests.
