## [Peer Review File · Nature Communications]

Formation of a single quasicrystal upon collision of multiple grainsREVIEWER COMMENTS

Reviewer #1 (Remarks to the Author):

The authors study the interplay between two growing quasicrystals with fixed seeds that might possess different orientations. MD simulations as well as XRT images from experiments are shown. As has been shown previously in PFC calculations, small differences in the orientation lead to the rotation of one of the quasicrystals and phasonic mismatches can be reduced by building up phasonic strain. The system and the results are important and interesting. However, the authors have to work out in a clearer way what is new about their study in order for the manuscript to be suitable for publication in Nature Communication. One possibility might be to explain in more detail how the results can be observed with the XRT study alone as this is (up to my knowledge) the first experimental observation of this process. In case the results cannot be obtained without the MD simulations, the authors have to explain in more detail how their results differ from the previous PFC calculations.

Related and additional comments:

- 1.) The authors claim that there are "several questions unanswered" due to the "limited resolution of the phase field crystal simulation". PFC calculation can resolve all particles. How can a MD simulation have a better resolution? Furthermore, what are the questions that have remained unanswered?
- 2.) The limited color ranges used in the figures make it hard to see some features. For example, in Fig. 1 the time sequence is hard to follow as the difference between subsequent images can be hardly seen. The color ranges in Figs. 2 and 3 should also be extended.
- 3.) In the MD results usually the left quasicrystal seems to win. Is there a reason for this bias?
- 4.) The authors might consider to compare the growth of periodic and aperiodic structures directly in the main paper instead of referring to the supplementary information. Furthermore, the number of dislocation that seems to be a suitable way to determine the critical angle might also be shown in the main article.
- 5.) The term "phasonic defect" might be misleading and should be avoided. Phasons are excitations and not really defects. On the other hand, defects like dislocations might contain a phasonic component. The authors should make sure to differentiate between phasonic excitations (i.e. phasonic strain) and dislocations.
- 6.) The authors claim the importance of their work for the control of defects and phasonic strain in the growth of quasicrystals. They should add some references on how phasons or defects affect the growth of quasicrystals and explain how their work compares to these references.
- 7.) The authors do not study how the critical angle depends on physical parameters, e.g., the temperature, which is fine. However, from the results and the physical of how the grains rotate, what kind of dependence do the authors expect?

Reviewer #2 (Remarks to the Author):

The authors show, through both experiment and simulation, the surprising result that decagonal quasicrystals, having their periodic axes nearly parallel and grown from fixed seeds, can rotate to merge and become a single large crystal with no grain boundary. Seed crystals which have their axes more mis-aligned do not rotate but rather keep their initial orientations and form clear grain boundaries. A mechanism for this is also proposed.

This work is certainly original and I find it convincing. I recommend publishing this paper.

Reviewer #3 (Remarks to the Author):

The paper describes novel in-situ experiments of grain coalescence and sintering of faceted, 100+ micrometer-scale, decagonal quasicrystal rods during solidification as a function of grain misorientation, and MD simulations of sintering and growth of nanometer-scale quasicrystal particles, where the initial number of atoms is 144 in each seed particle. I believe that the paper will be of significant interest to the community and to the wider field of interface motion and sintering, extending to other systems between classic FCC and quasicrystals. This may be particularly important for particle-particles systems when region between the two interacting grains is an interface-stabilized phase, known as a "complexion." This paper naturally leads to asking questions about how to use crystalline phase field modeling to capture the atomistics of what the authors observe at the large grain level through in-situ imaging and link it to their MD simulations. The level of detail and analysis, including the supplementary material, make this fertile ground for researchers to further explore the implications of the results.

There is only one concern that I believe the authors can easily remedy. There is a large difference in scale between the experimental results and the MD simulations. Will the authors provide justification for why MD, showing free rotation of non-faceted nanometer-sized particles, is relevant to rotation of 50 micrometer diameter, millimeter-long rods that are nucleated at the external oxide and are thus likely partially constrained in their ability to rotate.

Response to Reviewer #1

“The authors study the interplay between two growing quasicrystals with fixed seeds that might possess different orientations. MD simulations as well as XRT images from experiments are shown. As has been shown previously in PFC calculations, small differences in the orientation lead to the rotation of one of the quasicrystals and phononic mismatches can be reduced by building up phasonic strain. The system and the results are important and interesting. However, the authors have to work out in a clearer way what is new about their study in order for the manuscript to be suitable for publication in Nature Communication. One possibility might be to explain in more detail how the results can be observed with the XRT study alone as this is (up to my knowledge) the first experimental observation of this process. In case the results cannot be obtained without the MD simulations, the authors have to explain in more detail how their results differ from the previous PFC calculations.

The authors claim that there are ‘several questions unanswered’ due to the ‘limited resolution of the phase field crystal simulation’. PFC calculation can resolve all particles. How can a MD simulation have a better resolution? Furthermore, what are the questions that have remained unanswered?”

PFC methods are *continuum* models utilizing a scaled density field $\psi(\vec{r}, t)$ to describe ensemble-averaged mass density. They have been used to model liquid-solid transformations at diffusive timescales and with a spatial resolution at atomic level (Wang *et al.*, *Acta Mater.*, **58**, 4, 2010). Solving for $\psi(\vec{r}, t)$, which ranges from 0 to 1, yields the density field map. While these density field maps have atomic resolution, the number of crystal density wave peaks does not correspond directly to the number of atoms (Trautt *et al.*, *Acta Mater.*, **60**, 19, 2012). On the other hand, MD methods are *discrete* – they can represent exact atomic position of each atom at each time step. This means we can track individual particle trajectories. Phason flips can occur as isolated flips, as a string of consecutive flips, and in some cases, as a series of correlated flips between multiple particles (Engel *et al.*, *Phys. Rev. B.*, **82**, 134206). This makes trajectory analysis a powerful tool to understand phason dynamics. Research on phason dynamics in 3D decagonal quasicrystals, however, is limited. Our simulations can be used to further analyze phason dynamics during solidification and grain growth. We plan to work on the phason dynamics in the near future.

Here, we are interested in the question of grain coalescence at fixed seed position, which has not been addressed anywhere else, to the best of our knowledge. This question can be further divided into two parts: (1) “How is *rotational symmetry* at the grain boundary restored during grain coalescence?” and (2) “How is *translational symmetry* between multiple grains recovered during grain coalescence?” In this manuscript, we demonstrate that multiple grains are crystallographically aligned by sliding, and that translational symmetry is restored when phonon strain is redistributed as phason strain. What remains unanswered is *how* phasons restore translational symmetry after rotation. This is a question that can be answered by analysis of particle trajectories in future works with MD simulations, as mentioned above. Importantly, a fixed seed position is the norm in experiments, wherein heterogeneous nucleation prevails. Indeed, as the reviewer notes, we offer the first experimental confirmation of QC coalescence at fixed seed position. When seed position is not fixed, grains can displace each other (Schmiedeberg *et al.*, *Phys. Rev. E.*, **96**, 012602). Grain displacement could introduce alternate mechanisms for grain coalescence, or even aid in restoring translational mismatches in the lattices, resulting in different phason dynamics than those observed in simulations with fixed seeds. In short, we believe that fixing seed position introduces a novel, but important constraint to our simulations, and that MD is the most suitable tool for future work on the underlying phason dynamics.

“The limited color ranges used in the figures make it hard to see some features. For example, in Fig. 1 the time sequence is hard to follow as the difference between subsequent images can be hardly seen. The color ranges in Figs. 2 and 3 should also be extended.”

We adjusted the color ranges and used different colormaps in Figs. 1-3 for clearer visualization.

“In the MD results usually the left quasicrystals seems to win. Is there a reason for this bias?”

The reason for this apparent bias is statistical. We have cases where the yellow-orange (formerly blue) side wins and also where both grains rotate by the same amount. We have replaced Fig. 4 with a different case to better represent the behavior of the simulations.

“The authors might consider to compare the growth of periodic and aperiodic structures directly in the main paper instead of referring to the supplementary information. Furthermore, the number of dislocation that seems to be a suitable way to determine the critical angle might also be shown in the main article.”

We added the periodic crystal simulation (Fig. 4) and dislocation analysis (Fig. 3) in the main article.

“The term ‘phasonic defect’ might be misleading and should be avoided. Phasons are excitations and not really defects. On the other hand, defects like dislocations might contain a phasonic component. The authors should make sure to differentiate between phasonic excitations (i.e. phasonic strain) and dislocations.”

We removed the term “phason defect” from the abstract. We also edited instances in the discussion where we discuss dislocations and phason flips as mechanisms for phonon strain redistribution. Therein, we clarify that, although defects can have phononic components, phason strain and defect are not synonymous.

“The authors claim the importance of their work for the control of defects and phasonic strain in the growth of quasicrystals. They should add some references on how phasons or defects affect the growth of quasicrystals and explain how their work compares to these references.”

We demonstrated that the dislocation reactions redistribute the phonon strain from lattice mismatches as phason strain through a dislocation annihilation mechanism. The phason strain will be relaxed after prolonged annealing in real experiments, which helps to obtain defect-free quasicrystals. To the latter point, Nagao *et al.* (*Phys. Rev. Lett.*, **115**, 075501, 2015) observed grain boundary migration through an ‘error-and-repair’ process, wherein phason strain is introduced and subsequently relaxed to generate ideal quasicrystalline order. Furthermore, Han *et al.* (*Acta Cryst. A*, **75**, 281, 2019) explained that the faster growth rate of quasicrystals compared to approximant periodic crystals is attributed to the presence of phasons strain incorporated by random attachment of clusters to the growth front. We included these references in the manuscript to place our work in the context of others.

“The authors do not study how the critical angle depends on physical parameters, e.g., the temperature, which is fine. However, from the results and the physical of how the grains rotate, what kind of dependence do the authors expect?”

We thank the reviewer for the stimulating question. We define the critical angle θ_{crit} as the maximum misorientation (θ) above which grains are no longer able to rotate to minimize θ . In order for rotation to occur, the driving pressure ($P_{||}$) must be large enough to overcome the frictional forces in the system. As a result, θ_{crit} is determined by thermodynamic and kinetic factors, including the driving pressure and viscosity.

Following the reviewer’s suggestion, we first examine the effects of an increasing temperature on $P_{||}$, which contributes to increasing θ_{crit} . That is, a higher driving pressure should enable grain rotation to 0° misorientation even at higher θ . This can be understood as follows: $P_{||}$ for grain rotation is proportional to the first derivative of grain boundary energy (γ') as a function of θ (see, ①↔② in Fig. 1) and inversely proportional to the grain radius. More specifically, γ is proportional to a constant, γ_0 . This term depends on the rigidity modulus, lattice parameter, and Poisson’s ratio (Read and Shockley, *Phys. Rev.*, **78**, 275, 1950), which are all affected by temperature. Cheng *et al.* (*J. Appl. Phys.*, **123**, 085902, 2018) proposed recently a theoretical model for the temperature-dependent γ , in which γ decreases as temperature increases. Although this model

Figure 1: A schematic plot showing the relationship between the misorientation (θ) and grain boundary energy (γ) in the limit of low misorientation. The effect of θ ($\textcircled{1} \leftrightarrow \textcircled{2}$) and temperature changes ($\textcircled{1} \leftrightarrow \textcircled{3}$) on γ and the first derivative of γ with respect to misorientation (γ') can be found from the curves.

predicts temperature-dependence of γ , rather than γ' , it is based on temperature dependant factors in the constant γ_0 term. Since constants are maintained between derivatives, and the terms included in Cheng *et al.*'s model pertain to γ_0 , we expect γ' to decrease as a function of temperature as well. In Fig. 1, we schematically demonstrate how γ' varies with the magnitude of γ . Based on this finding, we expect γ' would decrease at high temperature ($\textcircled{1} \rightarrow \textcircled{3}$), along with the magnitude of γ , for fixed misorientation θ .

We examine also temperature-dependent external factors that impact P_{\parallel} , such as viscosity of liquid and presence of a premelting layer between grains. As temperature increases, the former generally decreases and the latter plays the role of lubricant between grains. Thus, the friction between multiple grains becomes smaller at higher temperature if there is no change in crystal geometry. In other words, we would require a lower P_{\parallel} to overcome the lower external friction at higher temperatures, and grain rotation can occur with lower driving pressure. Considering that *both* external friction and γ' decreases at higher temperatures, and they shift θ_{crit} in different directions, it is nontrivial to predict the temperature dependence of θ_{crit} .

Response to Reviewer #2

“The authors show, through both experiment and simulation, the surprising result that decagonal quasicrystals, having their periodic axes nearly parallel and grown from fixed seeds, can rotate to merge and become a single large crystal with no grain boundary. Seed crystals which have their axes more mis-aligned do not rotate but rather keep their initial orientations and form clear grain boundaries. A mechanism for this is also proposed.

This work is certainly original and I find it convincing. I recommend publishing this paper.”

We thank the reviewer for recognizing the novelty of our work.

Response to Reviewer #3

“The paper describes novel in-situ experiments of grain coalescence and sintering of faceted, 100+ micrometer-scale, decagonal quasicrystal rods during solidification as a function of grain misorientation, and MD simulations of sintering and growth of nanometer-scale quasicrystal particles, where the initial number of atoms is 144 in each seed particle. I believe that the paper will be of significant interest to the community and to the wider field of interface motion and sintering, extending to other systems between classic FCC and quasicrystals. This may be particularly important for particle-particles systems when region between the two interacting grains is an interface-stabilized phase, known as a “complexion.” This paper naturally leads to asking questions about how to use crystalline phase field modeling to capture the atomistics of what the authors observe at the large grain level through in-situ imaging and link it to their MD simulations. The level of detail and analysis, including the supplementary material, make this fertile ground for researchers to further explore the implications of the results.

There is only one concern that I believe the authors can easily remedy. There is a large difference in scale between the experimental results and the MD simulations. Will the authors provide justification for why MD, showing free rotation of non-faceted nanometer-sized particles, is relevant to rotation of 50 micrometer diameter, millimeter-long rods that are nucleated at the external oxide and are thus likely partially constrained in their ability to rotate.”

We are grateful that the reviewer acknowledges the potential impact of our work.

In our MD simulation, we fixed the seed positions and orientations to mimic the initial experimental conditions (*e.g.*, anchored quasicrystals with small misorientation, see Fig. 2). In spite of the fixed seeds, we were able to observe grain rotation and coalescence in both XRT experiment and MD simulation. According to existing theories, including those of Read and Shockley (*Phys. Rev.*, **78**, 275, 1950), Cahn and Taylor (*Acta Mater.*, **52**, 4887, 2004), and Trautt and Mishin (*Acta Mater.*, **60**, 2407, 2012), the driving pressure P_{\parallel} should be much smaller in the experiment because P_{\parallel} is inversely proportional to the grain radius, *i.e.*, larger grains will not be able to rotate as easily. However, it is still possible to rotate grains if the applied torque on grains is sufficient to overcome any interaction between grains or with an external substrate. To this end, there is ample experimental evidence showing grain rotation in unconstrained and fully-compact systems (R. Grupp *et al.*, *Nat. Comm.*, **2**, 298, 2011; J. M. Dake *et al.*, *Proc. Natl. Acad. Sci. U.S.A.*, **113**, E5998, 2016) and partly constrained grains with substrates (D. Heinen *et al.*, *J. Appl. Phys.*, **78**, 893, 1995). This suggests that the presence of external constraints can impact the kinetics of rotation; however, the thermodynamic driving force remains unchanged and grain rotation is still possible. We also observe grain rotation at experimental time-scales in the limit of small initial misorientation (see Figs. 1(a-b) of the manuscript). Our justification for the MD simulations is thus that they permit us to explore more comprehensively the preconditions (*i.e.*, misorientation) for grain coalescence.

REVIEWER COMMENTS

Reviewer #1 (Remarks to the Author):

Unfortunately, the reply by the authors is not sufficient. My major concern was (and still is) that there might not be enough new results to justify publication in Nature Communications. Note that the major results, e.g., the rotation of the grain and the proposed mechanism have already been studied and reported in [10]. As a consequence it is very important that the authors carefully explain what is new about their findings. According to their reply, the only new part of their work is that they fix their seed, which in my opinion is a shortage of their approach that they want to sell as a feature (see below for more details). No matter what you think about whether the seeds should be fixed or not, I do not think that this minor difference justifies the publication in Nature Communication.

As a consequence, given the reply, I cannot recommend publication of the manuscript. However, I strongly believe that the authors can do much better. In principle, there seem to be interesting simulation results and especially also experimental result. However, these results have to be analyzed more carefully and the authors have to explain how these results can provide a deeper insight into the system. It should not be the task of the reader (or the reviewer) to search for possible new results in the supplemental materials. Concerning the actual text of the manuscript at the moment, it remains unclear whether the experimental results can support the described mechanism at all (the author did not at all comment on the experiment in their reply). Furthermore, there is no discussion of the experimental results that significantly goes beyond the discussion in [10]. In fact, the analysis of phasonic rearrangements is more detailed in [10] while in the current manuscript and in the reply the authors only claim that an analysis is possible.

In my opinion the manuscript certainly has the potential to be published in Nature Communications, but only if the authors seriously work on the issues stated above.

Detailed comments:

- Fixed seeds vs. non-fixed seeds: I agree that fixed seeds might - in some details - lead to slightly different results than non-fixed seeds. However, these differences are not worked out in the manuscript. The results reported in the manuscript have also been observed in [10]. More importantly, I do not understand, why the seeds in the experiment should be assumed to be fixed. I somehow understand that due to the 3D-nature of the experiment the rotation of a seed might be difficult. However, I do not believe that a rearrangement of the seed by less than a typical distance between atoms is not possible in the experimental setup. How do you want to control the seed position against such small displacements? Finally, phasonic rearrangements certainly are possible even in the seed. In [10] it is shown that such phasonic rearrangements (also in the seeds) are essential to further lower the strain due to different orientations of the seeds. By neglecting this mechanism, the results in the submitted manuscript probably are probably biased because phasonic rearrangements are artificially suppressed in the seeds.

- Experiments: No reply was given on the experiments. My suggestion of my first report still is valid.

- Resolution of PFC models and analyses of phasonic dynamics: The authors provide a lengthy reply about possible PFC results in works with another context. However, the claim that PFC models cannot have an atomistic resolution is still wrong and I also strongly disagree with the claim that phasonic dynamics could not be analyzed in PFC models. I agree that phasonic flips in principle can be efficiently analyzed by using particle simulations. However, note that the authors do not provide such an analyzes (at least not as elaborated as in previous works). In contrast, in [10] some analyses of phasonic rearrangements is provided (within a PFC model) and note that in the papers cited in [10] (as well as papers citing [10]) even more elaborate analyses of phasonic dynamics and distortions in growing quasicrystals are given (with PFC calculations, particle simulations, or other approaches). The authors should cite some of these works and they should not claim that they could do better while they are actually not really doing it.

- Finally, I want to point out that the analysis provided in [10] on the maximum rotation angle depending on the distance of the seeds is more extensive than the somehow unspecific comments in the submitted manuscript.

Reviewer #3 (Remarks to the Author):

While the authors responded to the reviewers' comments in the Rebuttal Letter in some detail, they made few changes in the manuscript to clarify for future readers the reasons for their assertions. The purpose of providing reviews is not simply for reviewers to engage in a one-on-one dialogue with authors, but to point out issues with the manuscript. I do not agree with some of the interpretations of the literature provided in their rebuttal, but having those justifications explicitly articulated in the manuscript would allow readers to judge for themselves.

Below we address each concern point by point, in green text. Edits made to the manuscript in the first round of revisions are in red text, and those made in the second round are in blue text.

Response to Reviewer #1

Unfortunately, the reply by the authors is not sufficient. My major concern was (and still is) that there might not be enough new results to justify publication in Nature Communications. Note that the major results, e.g., the rotation of the grain and the proposed mechanism have already been studied and reported in [10]. As a consequence it is very important that the authors carefully explain what is new about their findings. According to their reply, the only new part of their work is that they fix their seed, which in my opinion is a shortage of their approach that they want to sell as a feature (see below for more details). No matter what you think about whether the seeds should be fixed or not, I do not think that this minor difference justifies the publication in Nature Communication.

As a consequence, given the reply, I cannot recommend publication of the manuscript. However, I strongly believe that the authors can do much better. In principle, there seem to be interesting simulation results and especially also experimental result. However, these results have to be analyzed more carefully and the authors have to explain how these results can provide a deeper insight into the system. It should not be the task of the reader (or the reviewer) to search for possible new results in the supplemental materials. Concerning the actual text of the manuscript at the moment, it remains unclear whether the experimental results can support the described mechanism at all (the author did not at all comment on the experiment in their reply). Furthermore, there is no discussion of the experimental results that significantly goes beyond the discussion in [10]. In fact, the analysis of phasonic rearrangements is more detailed in [10] while in the current manuscript and in the reply the authors only claim that an analysis is possible.

In my opinion the manuscript certainly has the potential to be published in Nature Communications, but only if the authors seriously work on the issues stated above.

We would like to stress the following two points regarding the impact of our work:

- To the best of our knowledge, our experimental results are the first to show grain coalescence in d-QCs (and perhaps among the first to show coalescence in real-time and 3D in *any* system). Although simulations like PFC or MD are a powerful tool to understand these and other systems, they have limitations in how accurately they can model reality. Our experimental results validate both our MD results and those PFC simulations of Schmiedeberg *et al.* (*Phys. Rev. E*, **96**, 012602, 2017). The discovery of this phenomenon in experiment is, in itself, novel and significant. It is facilitated by new developments in synchrotron-based 3D metrology that enable us to visualize the coalescence process in real-time. Only through nondestructive characterization can we

capture the morphologies, misorientations, and growth dynamics of the QCs.

Following the reviewer's constructive feedback, we now explicitly mention the novelty of our experiments on p. 2 of our revision (see blue text): "X-ray tomography [...] has opened a paradigm shift in solidification science, allowing us to capture transient microstructural dynamics in optically opaque materials [15]. To our advantage, it offers a relatively large (~1 mm) field-of-view together with a high localization precision (~600 nm), thus enabling us to screen materials at high-throughput and discover potentially rare events, such as QC coalescence in our work. To the best of our knowledge, our *in situ* experiments provide the first-ever evidence of this phenomenon." As an added benefit of 3D characterization, dihedral angles can be measured with high precision. As we now write on p. 3 of our revised manuscript, "[...] it is notoriously difficult to measure the groove angle ϕ (or *any* dihedral angle, for that matter) and track its evolution from a 2D section, unless the section is perfectly aligned to the aperiodic plane. With the aid of 3D characterization, we can circumvent this issue entirely." Lastly, we emphasize also that our experimental observations pertain to a statistically rare event, which we might not have been able to visualize through other characterization techniques, see the new paragraph on p. 3-4 of the manuscript as well as Fig. S3.

- We identify the mechanism of grain coalescence with the aid of MD simulations and reconcile our observations with classical theories. Ultimately, we find that QC grains rotate *via* grain boundary sliding (uncoupled to grain boundary migration). More succinctly, "QCs eliminate GBs by pure rotation," as we write on p. 9. Although Schmiedeberg *et al.* (*Phys. Rev. E*, **96**, 012602, 2017) do mention grain "rotation" and "displacement," it is unclear if they are necessarily coupled and/or prerequisite for coalescence. As we write in the revised text (p. 2), "Despite the insights obtained by Schmiedeberg *et al.* [10], fundamental questions remain on the mechanism of grain coalescence in QCs. Does it occur through grain rotation, GB translation, or a coupling between these two dynamic processes? Do these classical mechanisms apply when two QCs grow into stable structures before the ordered regions meet? What is the operant driving force?" These questions are answered on p. 8 of the manuscript, wherein we interpret our results according to classical theories (J. W. Cahn and J. E. Taylor, *Acta Mater.*, **52**, 4887, 2004; Z. T. Trautt and Y. Mishin, *Acta Mater.*, **60**, 2407, 2012). In accordance with theoretical predictions, we find that the "sliding velocity decreases with increasing misorientation between QC grains" (p. 8). By evaluating the suitability of these theories, we can place QCs on the same plane of analysis as their periodic counterparts, at least with respect to their kinetic properties. This is an important finding.

The reviewer also asked how the results can be observed with the XRT study alone and how we can correlate the experimental results with our proposed mechanism. To this end, we have added clarifying details pertaining to our experimental measurements on p. 2-3: "GBs may be detected in solidification experiments [26-28] owing to the fact that they create macroscopic depressions (grooves) of the solid-liquid interface around the point at which they emerge into the liquid." According to Young's law (see p. 3), the geometry of the groove (*i.e.*, the dihedral

angle) is directly related to the grain boundary energy, and hence the dislocation energy. By monitoring the dihedral angle as a function of time, we can assess if the GB is stable and fixed to the groove. Instead, we observe the annihilation of the groove, and hence the GB, following impingement. Taken altogether, our *in situ* data provides evidence of coalescence at low misorientation. Worth mentioning also is that “the groove itself does not move laterally along the solid-liquid interfaces on the intermediate time-scales (see arrow in Fig. 1(b)), which would indicate that GB translation on its own is not the mechanism of coalescence” (p. 3).

Since we visualize only the external solid-liquid interfaces *via* XRT, we turn to MD simulations to peer *inside* the QCs as they grow and coalesce. Furthermore, as we note on p. 4, “it is nearly impossible to preserve the QC interfaces at room temperature for high-resolution imaging, since the decagonal phase undergoes a peritectic decomposition.”

Detailed comments:

- Fixed seeds vs. non-fixed seeds: I agree that fixed seeds might - in some details - lead to slightly different results than non-fixed seeds. However, these differences are not worked out in the manuscript. The results reported in the manuscript have also been observed in [10]. More importantly, I do not understand, why the seeds in the experiment should be assumed to be fixed. I somehow understand that due to the 3D-nature of the experiment the rotation of a seed might be difficult. However, I do not believe that a rearrangement of the seed by less than a typical distance between atoms is not possible in the experimental setup. How do you want to control the seed position against such small displacements? Finally, phasonic rearrangements certainly are possible even in the seed. In [10] it is shown that such phasonic rearrangements (also in the seeds) are essential to further lower the strain due to different orientations of the seeds. By neglecting this mechanism, the results in the submitted manuscript probably are probably biased because phasonic rearrangements are artificially suppressed in the seeds.

Fixed seeds are dominantly observed in metal solidification wherein nucleation occurs heterogeneously. At our experimental resolution (600 nm), we do not detect the motion of seeds. As we now indicate in our revision (p. 2), seed displacement “would manifest itself as negative solid-liquid interfacial velocities on one side of the grain and positive interfacial velocities on the other. We do not observe this feature here nor in our prior experiments [22].” Here, positive velocities indicate growth (into the liquid) and negative velocities the converse.

We also note that the QC grains observed *via* XRT (Fig. 1) are nearly equivalent in size and also significantly larger than the critical seed size. Since the goal of our simulation is to model the experimental system, it is important that grains in our simulation are also nearly equivalent in size and sufficiently large. Although generally our MD simulations are on smaller length scales than our experiments, our additional simulations (Fig. S5(b)) with unfixed seeds suggest that the grains in our system are sufficiently large upon impingement. By “unfixed” we mean that particles in the seed are allowed to move freely following nucleation. In these new simulations of unfixed seeds, we saw some minor, random translational motion when grains were close to the

critical seed size. During the later stages of growth, and upon impingement, we do not see displacement of either grain. Comparing the later stages of growth in the fixed (Fig. S5(a)) and unfixed (Fig. S5(b)) cases, we see little difference in the crystal growth dynamics. This suggests that fixing seeds does not artificially suppress grain motion, and strengthens the assumption that our simulations can accurately model the phenomena observed in experiment. These details are now mentioned verbatim on p. 4 of the revision.

In contrast, the work by Schmiedeberg et al. (*Phys. Rev. E*, **96**, 012602, 2017) shows coalescence in a system with a relatively large grain size disparity, where one seed is slightly larger than the critical seed size before impingement (they write, “the smaller seed is only slightly larger than the critical seed size that is necessary to grow an ordered solid”). In their case, the difference in size between the two QC grains is therefore much more pronounced. Ultimately, we are operating in different limits: Schmiedeberg et al.’s study examines coalescence at early stages of growth, following nucleation, whereas we examine coalescence at the later stages during the growth process of two stable structures.

Reviewer #1 also noted that phasonic rearrangements can occur in the seed. While we agree with this statement in principle, we confirmed that restricting seed motion in our simulations does not have a significant impact on the mechanism driving coalescence (*i.e.*, constrained phasonic motion in the bulk of the crystal). Firstly, the seed makes up a very small portion of the crystal: Each seed contains 144 atoms, while each simulation has around 500,000 atoms total. That is, the seeds make up less than 0.06% of the atoms in the simulation. Secondly, the assumption that the fixed seed(s) are too small to cause an artificial suppression of phasonic motion in the bulk crystal appears valid in light of some of our preliminary particle trajectory analysis in Fig. S4. We cannot find any anomaly in density of particles showing particle flips around the fixed seeds. Rather, the number density of particles that flip are relatively uniform across the simulation domain, in simulations with fixed and unfixed seeds.

To summarize, although reviewer #1 raised valid concerns regarding our use of a fixed seed, additional analysis and simulations suggests that the decision to fix the seed in our simulations does not have any significant impact on the physics of grain coalescence in our system and therefore our MD simulations provide a good representation of the experimental system. We have added these analyses in the SI (namely, Figs. S4-S5) and addressed the impact of fixed seeds in our manuscript (p. 4), and thank the reviewer for their thoroughness.

- Experiments: No reply was given on the experiments. My suggestion of my first report still is valid.

We have since expanded the description of our experiments to emphasize the novelty of our contribution, see p. 2-3 and also our response to the first of the reviewer’s points.

- Resolution of PFC models and analyses of phasonic dynamics: The authors provide a lengthy reply about possible PFC results in works with another context. However, the claim that PFC models cannot have an atomistic resolution is still wrong and I also strongly disagree with the

claim that phasonic dynamics could not be analyzed in PFC models. I agree that phasonic flips in principle can be efficiently analyzed by using particle simulations. However, note that the authors do not provide such an analysis (at least not as elaborated as in previous works). In contrast, in [10] some analyses of phasonic rearrangements is provided (within a PFC model) and note that in the papers cited in [10] (as well as papers citing [10]) even more elaborate analyses of phasonic dynamics and distortions in growing quasicrystals are given (with PFC calculations, particle simulations, or other approaches). The authors should cite some of these works and they should not claim that they could do better while they are actually not really doing it.

Indeed, previous studies describe phason dynamics using PFC models in atomic resolution (J. Rottler *et al.*, *J. Phys.: Condens. Matter* **24**, 135002, 2012; C. V. Achim *et al.*, *Phys. Rev. Lett.*, **112**, 255501, 2014). We also agree that our assertion about the advantages of MD for trajectory analysis is a moot point when we do not include such analysis in our current manuscript (with the exception of the new Fig. S4). We acknowledge that our comments about the use of MD for an analysis of phason dynamics were too preemptive and we have now removed the relevant statements in the manuscript. We have since reframed our introductory paragraphs (see also our response to the reviewer's first point). Our goal was never to imply that we did better than other works on this point, but to complement and build on the works of others. Specifically, we sought to conduct an in-depth analysis of the mechanism behind grain coalescence, and compare it to classical theories of grain coalescence, which, to the best of our knowledge, has not been done previously for quasicrystals. The novelty and significance of this effort were covered previously in this response letter, and the manuscript has been edited accordingly.

- Finally, I want to point out that the analysis provided in [10] on the maximum rotation angle depending on the distance of the seeds is more extensive than the somehow unspecific comments in the submitted manuscript.

By fixing the distance between seeds, we can readily determine the maximum rotation angle θ_{crit} since the grain size upon merging is almost constant, along with other competing factors, throughout the entire set of MD simulations. Thus, we can eliminate the influence of grain size on the dynamics of coalescence (note the sliding velocity is inversely proportional to the grain size, as mentioned on p. 8). That said, as we noted in our previous response letter, θ_{crit} depends on other materials properties that vary with alloy composition. Unfortunately, AlCoNi decagonal QCs have not yet been assembled from atomistic pair potentials. Although our MD model accurately captures the 3D atomic structure of the experimental system, our simulation is limited in its ability to accurately model composition-dependent properties. Since the primary goal of our simulations is to describe the mechanisms that give rise to the behaviour seen experimentally, we would like to refrain from making specific comments regarding the exact value of θ_{crit} , as it may not pertain to our experimental system.

Response to Reviewer #3

While the authors responded to the reviewers' comments in the Rebuttal Letter in some detail, they made few changes in the manuscript to clarify for future readers the reasons for their assertions. The purpose of providing reviews is not simply for reviewers to engage in a one-on-one dialogue with authors, but to point out issues with the manuscript. I do not agree with some of the interpretations of the literature provided in their rebuttal, but having those justifications explicitly articulated in the manuscript would allow readers to judge for themselves.

We appreciate your advice. We have since added the details of our discussion in the manuscript itself, on p. 8: “[...] grains may rotate if the applied torque is sufficient to overcome any interaction between grains or with an external substrate. To this end, there is ample experimental evidence showing grain rotation in unconstrained and fully compact systems [36,37] and partially constrained grains with substrates [38]. This suggests that the presence of external constraints may impact the kinetics of rotation; nevertheless, grain rotation is still possible. A case in point is Figs. 1(a-b) from our own work, showing grain rotation on experimental time-scales in the limit of small initial misorientation.”

REVIEWER COMMENTS

Reviewer #1 (Remarks to the Author):

The reply letter of the authors is again much more elaborate than the (minor) changes in the manuscript. Furthermore, the authors unfortunately do not follow my suggestions (not in the reply letter and not in the revised manuscript).

So the main question is: Are there enough new results to justify publication in Nature Communications?

In my opinion, most of the reasons given by the authors do not justify publication in Nature Communications. I will comment on them at the end of this report.

However, there are reasons why the manuscript might be made suitable for Nature Communications:

- Obviously, the experimental results are fascinating. Unfortunately, the main text of the manuscript concentrates on the results of the simulations while most experimental results are only shown in the supplemental materials. The discussion of these experimental results is very short almost suggesting that one does not learn anything from this experiment (which in my opinion is not the case).

- Fig. S6 shows the main result of the experiment (grain rotation for small mismatch angles but no grain rotation for larger angles) that can be used as a justification for publication.

- A comparison of Fig. S7 with the simulation results would be a major advancement.

- The phason relaxation in S11 probably is the most spectacular result! If analyzed and discussed in a little bit more detail this might demonstrate how the phasonic relaxation in quasicrystals can be associated to self-healing, a buzzword that the authors use in their title but hardly discuss in the text (and most important: In their present discussion the authors fail to demonstrate why self-healing in quasicrystals might be different from periodic systems).

- Phasonic strain and phasonic rearrangements in the simulations are accessible and should be discussed.

Concerning the major reasons for publication as given by the authors:

- In the first round the authors claimed that molecular dynamics simulations are superior to PFC calculations in case one wants to track particle positions and that the use of PFC was a shortcoming of a previous study. Fortunately, the authors do no longer stick to this argument and they seem to agree that PFC calculations are not as bad as they have claimed.

- In the second round the authors claimed that fixed seeds are superior to non-fixed seeds. In contrast, the second reply letter of the authors reads like a defense of their choice and why it is not a complete failure to assume fixed seeds. Given the size of their system, I agree that differences between fixed and non-fixed seeds probably are small. However, fixed seeds should not be sold as a major advantage of the manuscript.

- In the third round the authors claim that grain boundary motion was not ruled out in a previous work and that the merger of stable regions was not studied. Both claims are incorrect in my opinion though the previous study indisputable had a different focus namely on the difference between aperiodic and periodic structures. At the moment such differences are not really discussed in the present manuscript but questions are asked that do not really depend on whether the system is periodic or aperiodic. So concerning the discussion of the system, the present manuscript seems to be a step back in comparison to previous results.

In conclusion, I suggest that the authors look for what the strength of their work is without trying

to diminish previous results or other approaches (PFC, non-fixed seed, study of phasons). It seems not to be the intention to diminish the work of others but somehow they do not succeed to make their point in another way. If there are no significant differences from the results reported previously that is fine as well; there already is a major advancement concerning the methods (e.g., experimental results!).

Minor issues:

- In Fig. S5 the interfaces seem to meet at very different times. Probably the distance between the seeds is different such that the shown examples cannot be compared. The authors should show cases that are comparable.

- Fig. S5, caption, I do not understand the second sentence.

- Fig. S3 seems to demonstrate that the cases shown in the manuscript are very unlikely and thus irrelevant. I do not think that this is what the authors want to show.

Reviewer #3 (Remarks to the Author):

The title does not describe the subject of the paper. This paper describes the sintering of two d-QC rods. Using the terms "Self-healing" and "hard collisions" does not make sense. The following paragraph is a misstatement of the relevance and impact of their work: "8 single d-QC from multiple grains. This experimental technique has opened a paradigm shift in solidification science, allowing us to capture transient microstructural dynamics in optically opaque materials [15]. To our advantage, it offers a relatively large (~1 mm) field-of-view together with a high localization precision (~600 nm), thus enabling us to screen materials at high-throughput and discover potentially rare events, such as QC coalescence in our work. To the best of our knowledge, our in situ experiments provide the first-ever evidence of this phenomenon." Other people have been using this technique for quite some time, even for solidification research. Yes, they used this technique to perform measurements of the sintering of d-QC rods. To claim that it is "high-throughput" is not justified. They talk about the sintering kinetics and mechanisms for a pair of rods. Saying that their results are "first ever" may be true, but not ground breaking. This discussion of dihedral angles is verbose. The authors comments about heterogeneous nucleation requiring fixed seeds is incorrect. Heterogeneous nucleation requires a substrate or particle on which nucleation is easier. There is no requirement that those be fixed in space. As early as the 1950's the classic droplet solidification experiments from Turnbull demonstrated that. In the end, comparing experimental coarsening/sintering results from hundreds of micrometer long d-QC rods with 100 micrometer diameters with simulations of two particles with a few hundred atoms is a stretch. This paper could be condensed considerably.

Below we address each concern point by point, in green text. New edits to the manuscript are marked in purple text. Edits made to the manuscript in the first round of revisions are retained in red text, and those made in the second round are in blue text.

Response to Reviewer #1

The reply letter of the authors is again much more elaborate than the (minor) changes in the manuscript. Furthermore, the authors unfortunately do not follow my suggestions (not in the reply letter and not in the revised manuscript).

So the main question is: Are there enough new results to justify publication in Nature Communications?

In my opinion, most of the reasons given by the authors do not justify publication in Nature Communications. I will comment on them at the end of this report.

Our main contribution is that we experimentally demonstrated the coalescence of metallic quasicrystals in 4D (*i.e.*, 3D space plus time), and for the first time. It is generally difficult to reconstruct the dynamics of grains and interfaces through any other means, *e.g.*, *post mortem* microscopy. Furthermore, our simulations support the *in situ* experimental results and previous studies by examining the underlying kinetics of grain coalescence in quasicrystals, *e.g.*, the contributions of phasons to the relaxation process (Fig. 7).

However, there are reasons why the manuscript might be made suitable for Nature Communications:

- Obviously, the experimental results are fascinating. Unfortunately, the main text of the manuscript concentrates on the results of the simulations while most experimental results are only shown in the supplemental materials. The discussion of these experimental results is very short almost suggesting that one does not learn anything from this experiment (which in my opinion is not the case).

We have followed the reviewer's suggestion by incorporating more experimental results, namely our analysis of interface orientations from X-ray tomography (see revised Fig. 1) and also phonon/phason strain from X-ray diffraction (see new Fig. 2). The experimental and simulation results complement each other (*cf.* Figs. 1 and 3) and make our argument even stronger. We thank the reviewer for helping us make our manuscript more balanced.

- Fig. S6 shows the main result of the experiment (grain rotation for small mismatch angles but no grain rotation for larger angles) that can be used as a justification for publication.

- A comparison of Fig. S7 with the simulation results would be a major advancement.

We agree that Figs. S6 and S7, which were included in our second-round revision, show "grain rotation for small mismatch angles but no grain rotation for larger angles" as noted by the reviewer. However, these figures are based on the MD simulations (please refer to the first line of both captions, which mention "simulation" explicitly).

Nevertheless, we appreciate the spirit of the reviewer's suggestion of a direct comparison between experiment and simulation. From our 3D X-ray imaging data, we can measure the orientations (normal vectors) of the solid-liquid interfaces; we can then project these normal vectors onto a stereographic grid. The resulting plot (so-called "interface normal distribution" or IND for short) conveys the directionality of the solid-liquid interface (here, at 910 °C). For a single d-QC, the IND should include ten peaks corresponding to the ten crystallographically-equivalent {10000} facets. If instead there exist multiple independent QC grains in the liquid, the IND is a superposition of multiple single crystal patterns. The IND is thus a convenient fingerprint of polycrystallinity.

To clarify the link between the experimental INDs and simulated diffraction patterns, we note that the external morphology of a crystal reflects its point group isogonal with its space group (M. J. Buerger, "Elementary Crystallography", John Wiley and Sons, 1956). This principle allows us to relate the IND from experiment and the diffraction patterns from simulation. To draw attention to this comparison, we decided to combine Fig. S2 with Fig. 1 and Fig. S6 with Fig. 3 and expand our discussion accordingly (see the new text starting with, "This is visible in the diffraction pattern ..." on p. 6 of the revised manuscript).

Furthermore, using one pair of two opposite Bragg peaks (aligned in the same direction) in the diffraction image shown in Fig. S6, we could derive Fig. S7, which brings to light the existence of dislocations in highly misoriented structures. We rely on this same approach in the new Fig. 4, which shows a magnified view of the region where two QCs collide.

- The phason relaxation in S11 probably is the most spectacular result! If analyzed and discussed in a little bit more detail this might demonstrate how the phasonic relaxation in quasicrystals can be associated to self-healing, a buzzword that the authors use in their title but hardly discuss in the text (and most important: In their present discussion the authors fail to demonstrate why self-healing in quasicrystals might be different from periodic systems).

Following the reviewer's suggestion, we combined Fig. S11 from the second revision with the X-ray diffraction pattern (formerly Fig. S1) into a new Fig. 2. This figure proves that phason strain relaxation occurs during slow cooling. We have also discussed this figure on p. 2 (see the new text beginning with "On the basis of our XRD results, ...") and refer again to it at the end of the manuscript on p. 9 ("In addition, we confirm the relaxation ...").

To avoid any confusion, we changed the title of our manuscript to "Formation of a single quasicrystal grain upon collision of multiple grains". We believe this new title is in line with our main finding (see also our response to the reviewer's first comment). To their last point, the distinguishing feature of QCs is the presence of phason strain, which we characterize in Fig. 7 and describe on p. 9-10 in relation to the mechanism of coalescence.

- Phasonic strain and phasonic rearrangements in the simulations are accessible and should be discussed.

We thank the reviewer for their suggestions. We moved our phason strain measurements from the SI (formerly Fig. S10) to the manuscript (now Fig. 7) and expanded our discussion of phason strain in simulation (see text starting with "The density modes ..." on p. 10). We

demonstrated that higher phason strain remains at the grain boundary region, especially in cases where two quasicrystals with higher initial misorientation coalesce.

Our manuscript includes phasonic rearrangement analysis to justify our use of fixed seed simulations. While we acknowledge that our discussion of phasonic rearrangements is limited, we feel that extensive trajectory analysis is beyond the scope of this work, as noted on p. 2 of the SI. We reserve further analysis for future studies.

Concerning the major reasons for publication as given by the authors:

- In the first round the authors claimed that molecular dynamics simulations are superior to PFC calculations in case one wants to track particle positions and that the use of PFC was a shortcoming of a previous study. Fortunately, the authors do no longer stick to this argument and they seem to agree that PFC calculations are not as bad as they have claimed.

Indeed. Thanks for understanding our intention.

- In the second round the authors claimed that fixed seeds are superior to non-fixed seeds. In contrast, the second reply letter of the authors reads like a defense of their choice and why it is not a complete failure to assume fixed seeds. Given the size of their system, I agree that differences between fixed and non-fixed seeds probably are small. However, fixed seeds should not be sold as a major advantage of the manuscript.

The main reason for choosing fixed seeds is that we do not detect seed displacement during the experiments. We thus intended to simulate “anchored” decagonal quasicrystals to follow the condition that we experimentally observed. As we demonstrated in the second round of revisions, the fixed seed simulation does not cause any significant difference compared with the non-fixed simulation. Therefore, we agree with the reviewer that simulation with fixed seeds is not the major advantage of the manuscript.

- In the third round the authors claim that grain boundary motion was not ruled out in a previous work and that the merger of stable regions was not studied. Both claims are incorrect in my opinion though the previous study indisputable had a different focus namely on the difference between aperiodic and periodic structures. At the moment such differences are not really discussed in the present manuscript but questions are asked that do not really depend on whether the system is periodic or aperiodic. So concerning the discussion of the system, the present manuscript seems to be a step back in comparison to previous results.

The reviewer’s comments have helped us realize that the way we posed our questions misrepresented Schmiedeberg et al.’s work. We apologize for implying that previous work did not rule out grain boundary motion — this was unintentional. After careful consideration of the reviewer’s comments, we have reframed our questions to properly reflect the novelty of our work (*i.e.*, the experimental results). See on p. 2 the revised question, “Can coalescence take place in experimental systems?”

In addition, we agree that the stability of grains was discussed in their work. When the distance between two seeds is far away from each other, two seeds can evolve into stable quasicrystals. Consequently, they form a grain boundary in between.

Lastly, the difference between aperiodic and periodic structures are discussed in our manuscript. In Fig. 6, grain coalescence was not observed between FCC grains after grain rotation by sliding, which indicates that the incommensurate lattice structure plays a key role in grain coalescence (see also corresponding discussion on p. 9-10 and Fig. 7). To summarize, in Fig. 7, we observe that phason density increases with initial grain misorientation. On p. 9-10 (beginning with “Due to the unique symmetries of QCs, ...”), we discuss how QC dislocations contain both phononic and phasonic components, whereas defects in periodic crystals only contain phononic components. This indicates that QCs can eliminate translational mismatches in the lattice through dislocation reactions that transform phonon strain into phason strain. Since phasons arise due to the aperiodicity and unique symmetries of QCs (discussed in detail on p. 1 and mentioned in p. 9), we feel that the plausibility of grain coalescence following grain rotation does indeed depend on “whether the system is periodic or aperiodic,” to quote the reviewer.

In conclusion, I suggest that the authors look for what the strength of their work is without trying to diminish previous results or other approaches (PFC, non-fixed seed, study of phasons). It seems not to be the intention to diminish the work of others but somehow they do not succeed to make their point in another way. If there are no significant differences from the results reported previously that is fine as well; there already is a major advancement concerning the methods (e.g., experimental results!).

We updated our manuscript not to diminish the work of others but rather to emphasize our major findings, see p. 2-3. We thank you again for acknowledging our experimental work.

Minor issues:

- In Fig. S5 the interfaces seem to meet at very different times. Probably the distance between the seeds is different such that the shown examples cannot be compared. The authors should show cases that are comparable.

The unfixed seeds and the fixed seeds had a misorientation of 9° and a distance of $40d$ (where $d = 1.02$, or the location of the first local minimum in the pair potential). The difference between collision times between the two simulations is likely stochastic. The caption has been updated to clarify the fact that these examples are comparable in distance, and to address the issue mentioned in the reviewer's comments below.

- Fig. S5, caption, I do not understand the second sentence.

We have updated the caption for Fig. S5 and hope it is clearer.

- Fig. S3 seems to demonstrate that the cases shown in the manuscript are very unlikely and thus irrelevant. I do not think that this is what the authors want to show.

Thanks for pointing this out. We have removed Fig. S3.

Response to Reviewer #3

The title does not describe the subject of the paper. This paper describes the sintering of two d-QC rods. Using the terms "Self-healing" and "hard collisions" does not make sense.

We changed the title to "Formation of a single quasicrystal grain upon collision of multiple grains" for a more intuitive description. The term "collision" is used in a number of solidification textbooks (e.g., J. A. Dantzig and M. Rappaz, *Solidification*, EPFL Press, 2009) to indicate the interaction between multiple grains in a liquid.

The following paragraph is a misstatement of the relevance and impact of their work: "single d-QC from multiple grains. This experimental technique has opened a paradigm shift in solidification science, allowing us to capture transient microstructural dynamics in optically opaque materials [15]. To our advantage, it offers a relatively large (~1 mm) field-of-view together with a high localization precision (~600 nm), thus enabling us to screen materials at high-throughput and discover potentially rare events, such as QC coalescence in our work. To the best of our knowledge, our in situ experiments provide the first-ever evidence of this phenomenon."

Other people have been using this technique for quite some time, even for solidification research. Yes, they used this technique to perform measurements of the sintering of d-QC rods. To claim that it is "high-throughput" is not justified. They talk about the sintering kinetics and mechanisms for a pair of rods. Saying that their results are "first ever" may be true, but not ground breaking.

We are sorry for the confusion. Indeed, X-ray tomography (XRT) and its variants have been used widely for solidification research (L. Arnberg and R. H. Mathiesen, *JOM* **59**, 20, 2007; S. Akamatsu and H. Nguyen-Thi, *Acta Mater.* **108**, 325, 2016; A. J. Shahani *et al.*, *Mater. Res. Lett.*, **8**, 462, 2020). We now reference these contributions on p. 2. Here, we utilized this technique to detect the formation of a single quasicrystals from multiple grains, for the first time. Our intention to use the word, "high-throughput" was to emphasize the large datasets outputted from XRT, however, this may sound arbitrary. We have since removed the referenced sentence from the manuscript.

This discussion of dihedral angles is verbose.

We took out this description that we had added in the second round revision.

The authors comments about heterogeneous nucleation requiring fixed seeds is incorrect. Heterogeneous nucleation requires a substrate or particle on which nucleation is easier. There is no requirement that those be fixed in space. As early as the 1950's the classic droplet solidification experiments from Turnbull demonstrated that.

We agree with your comment and have rewritten the referenced sentence. In general, heterogeneous nucleation may occur on a fixed substrate (e.g., oxide skin in our case) or on floating particles (e.g., grain refiners, atomic clusters suspended in the liquid).

In our case, we observe that there is no displacement of d-QCs in the XRT experiment owing to the fact that the grains nucleated heterogeneously on the fixed oxide layer.

In the end, comparing experimental coarsening/sintering results from hundreds of micrometer long d-QC rods with 100 micrometer diameters with simulations of two particles with a few hundred atoms is a stretch.

We agree that the size of our simulations is smaller than our experiments. Simulations are computationally too expensive to replicate the length scales observed in experiments. However, each simulation consists of approximately half a million atoms in 3D. Our simulations are large enough to capture the contributions of phasons to grain coalescence and their density as a function of misorientation, see p. 9-10 of the manuscript. We also note that molecular dynamics simulations of smaller or comparable size (Z. T. Trautt and Y. Mishin, *Acta Mater.* **60**, 2407, 2012; M. Upmanyu *et al.*, *Acta Mater.* **54**, 1707, 2006) have been used to successfully model equations of grain coalescence (albeit in periodic systems), which are derived based on symmetry and should be independent of system size.

This paper could be condensed considerably.

Thanks for your advice. We have condensed our manuscript by removing the following:

- Experimental details on synchrotron X-ray tomography
- Description of grain boundary energy and groove angle
- Phenomenological equations of grain rotation
- Description of results regarding the former Fig. S3

That said, we had to balance your feedback with that of the first reviewer, who asked us to analyze and discuss the experimental results “in a little bit more detail.” We have thus removed as much as possible, while still fulfilling the request of the first reviewer.